# A Near-Optimal Best-of-Both-Worlds Algorithm for Federated Bandits

**Zicheng Hu, Zihao Wang, Cheng Chen**[*]
East China Normal University
{51275902019, 10235101442}@stu.ecnu.edu.cn
chchen@sei.ecnu.edu.cn

## Abstract

This paper studies federated multi-armed bandit (MAB) problems in which multiple agents work together to solve a common MAB problem through a communication network. We focus on the heterogeneous setting in which no single agent can identify the globally best arm using only locally biased observations. In this setting, different agents may select the same arm at the same time step, but receive different rewards. We propose a novel algorithm called FEDFTRL for this problem and, to our knowledge, it is the first to achieve near-optimal regret guarantees in both stochastic and adversarial environments. Notably, in the adversarial regime, our algorithm achieves $O(T^{\frac{1}{2}})$ regret, a significant improvement over the state-of-the-art regret of $O(T^{\frac{2}{3}})$ (Yi & Vojnovic, 2023). We also provide empirical evaluations comparing our algorithm with baseline methods, demonstrating the effectiveness of our approach on both synthetic and real-world datasets.

## 1 Introduction

The multi-armed bandit (MAB) problem is one of the most fundamental settings in online learning. Motivated by the emerging paradigm of federated learning in which multiple heterogeneous agents collaboratively train a model without sharing their raw data (Kairouz et al., 2021), many recent studies have explored MAB problems in federated environments. In the federated bandit problem, the goal of all agents is to identify a globally best arm, and each agent can only observe locally biased rewards without sharing its raw data with others.

Federated bandits arise in many real-world scenarios where each agent's sequence of arm pulls and outcomes is kept private. For example, in a personalized online education system, optimizing a student's performance (i.e., rewards) often requires tailoring instructional methods (i.e., arms) to the student's individual characteristics (Cai et al., 2021). Given that educational software often operates locally on students' devices, it is essential for the central educational platform to personalize learning experiences effectively while maintaining strict privacy constraints. Specifically, the platform should adapt teaching strategies based on each student's unique context without collecting sensitive personal attributes or performance data.

In the federated bandit problem, there are $V$ agents, each selecting one of $K$ arms in each round. Each agent observes a heterogeneity-induced, locally biased reward for the chosen arm and communicates solely with its neighbors. The agents collectively aim to identify the globally best arm and maximize the group's cumulative reward, without sharing raw reward observations with one another. Prior works on federated bandits mainly focus on two settings: (i) *stochastic* settings (Dubey & Pentland, 2020; Zhu et al., 2021; Huang et al., 2021; Shi et al., 2021; Réda et al., 2022), where rewards are drawn from underlying distributions; and (ii) *adversarial* settings (Yi & Vojnovic, 2023), where rewards are arbitrarily chosen by an adversary. However, in practice, environments are seldom purely stochastic or fully adversarial, and their precise nature is often unknown. Despite this, the existing literature on federated bandits continues to adhere to the traditional distinction between stochastic and adversarial settings.

---

[*]The corresponding author

| Settings | Algorithms | Individual regret |
|---|---|---|
| Stochastic | Gossip_UCB (Zhu et al., 2021) | $O\big(\sum_{k \neq k^*} \frac{V \log T}{\Delta_k}\big)$ |
| | DRRB-bandit (Zhang et al., 2025) | $O\big(\sum_{k \neq k^*} \frac{\log T}{V \Delta_k}\big)$ |
| | FEDFTRL (Ours) | $O\big(\sum_{k \neq k^*} \frac{\log T}{V \Delta_k}\big)$ |
| | Lower bound (Zhu et al., 2021) | $\Omega\big(\sum_{k \neq k^*} \frac{\log T}{V \Delta_k}\big)$ |
| Adversarial | FEDEXP3 (Yi & Vojnovic, 2023) | $O\big(\sqrt{C_T^P \log K}\, K^{\frac{2}{3}} T^{\frac{2}{3}}\big)$ |
| | FEDFTRL (Ours) | $O\big(\sqrt{\frac{KT}{V}} + \sqrt{C_T^P T \log K}\big)$ |
| | Lower bound (Yi & Vojnovic, 2023) | $\Omega\big(\max\big\{\sqrt{\frac{KT}{V}}, \sqrt[4]{\frac{1+d_{\max}}{\lambda_{V-1}(M)}}\sqrt{T \log K}\big\}\big)$ |

Table 1: **Overview of best-known regret bounds for federated bandits.** Here, $P$ denotes the doubly stochastic communication matrix over the network $G$, and $\sigma_2(P)$ is its second-largest singular value. We define $C_T^P := \frac{\min\{\log(VT), \sqrt{V}\}}{1 - \sigma_2(P)} + 2 + D$, where $D$ is the diameter of $G$, capturing the dependence on the network topology. Let $M$ denote the Laplacian matrix of $G$; $\lambda_2(M)$ is its second-smallest eigenvalue, and $d_{\max}$ is the maximum node degree in $G$.

In this paper, we study *best-of-both-worlds (BOBW)* algorithms for federated bandits, i.e., methods that achieve near-optimal regret bounds in both stochastic and adversarial regimes. We propose a variant of the Follow-The-Regularized-Leader (FTRL) framework for federated bandits, which incorporates a novel communication scheme. Since each agent can only communicate with its neighbors in each round, information from agents beyond the immediate neighborhood is received only after multiple rounds. We regard the resulting latency as a form of feedback delay. Based on this idea, we develop our algorithm by adopting a hybrid regularizer (Zimmert & Seldin, 2020; Masoudian et al., 2022) for bandits with delayed feedback and introducing a novel learning-rate schedule. Additionally, to address the heterogeneity-induced feedback, we introduce a novel truncated loss estimator that ensures the action probabilities of each agent remain nearly aligned and keeps the aggregate loss estimator at each time step closer to the cross-agent average loss.

Another technical contribution of this work is a novel analysis of individual regret. Unlike prior studies on multi-agent bandits that directly analyze each agent's regret, we first establish an upper bound on the group regret. Leveraging the fact that agents' action probabilities are nearly aligned, we can translate the group-regret bound into per-agent regret bounds that scale approximately as $1/V$ (up to small error terms). This approach allows us to obtain near-optimal individual regret bounds in both stochastic and adversarial regimes.

To keep the presentation simple, we assume that there exists a unique best arm $k^*$. Our method can be generalized to settings with multiple best arms by leveraging techniques from Ito (2021b). The regret bounds of our method, along with comparisons to recent works, are presented in Table 1. Our contributions are summarized as follows:

- We provide an anytime near-optimal federated bandit algorithm, called FEDFTRL, which achieves an individual regret bound of $O\left(\sum_{k \neq k^*} \frac{\log T}{V \Delta_k} + \frac{C_T^P}{\Delta_k \log K}\right)$ in the stochastic regime and simultaneously achieves an individual regret bound of $O\left(\sqrt{\frac{KT}{V}} + \sqrt{C_T^P T \log K}\right)$ in the adversarial regime. Here, $C_T^P$, defined in equation 2, captures the topology of the communication graph. To our knowledge, FEDFTRL is the first method to achieve a BOBW regret guarantee in federated bandits, and its individual regret bounds match the corresponding lower bounds up to low-degree polynomial factors.

- In the adversarial regime, prior work (Yi & Vojnovic, 2023) achieves an $O(T^{\frac{2}{3}})$ regret bound, whereas our method achieves a tighter $O(T^{\frac{1}{2}})$ bound.

- We conduct experiments on both synthetic and real-world datasets to demonstrate the effectiveness of our method. The empirical results show that our algorithm significantly outperforms prior approaches.

## 2 RELATED WORK

**Federated bandits.** In the stochastic setting, Dubey & Pentland (2020) and Huang et al. (2021) studied linear contextual federated bandits and extended the classical LinUCB algorithm (Li et al., 2010) to the federated environment. Shi et al. (2021) formally defined the federated bandit problem and proposed an optimal algorithm for a centralized communication network. Zhu et al. (2021) were the first to study federated bandits in a decentralized network, applying efficient gossip-based communication to achieve a near-optimal regret bound. Recently, Zhang et al. (2025) proposed a fully distributed online consensus estimation approach and integrated it into a distributed successive elimination algorithm, achieving an optimal regret bound. In the adversarial setting, Yi & Vojnovic (2023) were the first to formalize federated bandits without stochastic assumptions on the rewards, termed the *doubly adversarial bandit* problem. They also proposed FEDEXP3, which achieves a sublinear regret of order $O(T^{\frac{2}{3}})$ in this setting.

**Best-of-Both-Worlds.** Traditionally, stochastic and adversarial environments have been studied independently. However, in practice, the nature of the environment is often unknown or may vary over time. This has motivated increasing interest in algorithms that perform well in both stochastic and adversarial settings, a paradigm commonly referred to as BOBW (Bubeck & Slivkins, 2012; Auer & Chiang, 2016; Seldin & Lugosi, 2017; Wei & Luo, 2018). Zimmert & Seldin (2021) applied a Tsallis-INF regularizer within the FTRL framework to achieve BOBW guarantees for stochastic and adversarial regimes, as well as a continuum of intermediate regimes. Leveraging FTRL's flexibility and strong theoretical properties, subsequent work has extended BOBW results to more complex settings, including combinatorial bandits (Zimmert et al., 2019; Ito, 2021a; Tsuchiya et al., 2023b), linear bandits (Lee et al., 2021; Dann et al., 2023), graph bandits (Rouyer et al., 2022; Ito et al., 2022), partial monitoring (Tsuchiya et al., 2023a), and delayed feedback (Masoudian et al., 2022). Among these, FTRL variants addressing delayed feedback are particularly relevant to our work, as federated bandits inherently involve implicit delays due to decentralized communication. Our algorithm builds on this line of research, adapting the FTRL paradigm to accommodate both heterogeneous rewards and decentralized communication while preserving a BOBW guarantee.

## 3 PRELIMINARIES

Let $[V] = \{1, 2, \ldots, V\}$ and $[K] = \{1, 2, \ldots, K\}$ denote the sets of agents and arms, respectively. The network of $V$ agents is represented by a simple, connected, undirected graph $G = ([V], E)$, where $E$ is the set of edges. The diameter $D$ is the maximum shortest-path distance between any pair of nodes in $G$.

We consider a heterogeneous multi-agent system in which all agents collaboratively solve a common $K$-armed bandit problem over a horizon of $T$ rounds. At each time step $t \in [T]$, each agent $v$ selects an arm $k_{v,t}$ according to its own strategy and then observes the locally biased loss of the chosen arm, $\ell_{v,t}(k_{v,t}) \in [0, 1]$, where $\ell_{v,t}(k)$ denotes the loss of arm $k$ for agent $v$ at time $t$. The *average loss* of arm $k$ at time $t$ is defined as the average loss across all agents:

$$\bar{\ell}_t(k) = \frac{1}{V} \sum_{v=1}^{V} \ell_{v,t}(k).$$

At the end of each time step $t$, each agent $v$ can exchange information with its neighbors $\mathcal{N}(v) = \{u \in [V] : (v, u) \in E\}$. The received information can be used in the next round if desired. The communication process is characterized by a matrix $P \in [0, 1]^{V \times V}$, where $P_{u,v} = 0$ whenever $u \neq v$ and $(u, v) \notin E$ (i.e., agents communicate only with their neighbors). We assume that $P$ is doubly stochastic, i.e.,

$$\sum_{u \in [V]} P_{u,v} = \sum_{v \in [V]} P_{u,v} = 1, \quad P_{u,v} \geq 0.$$

We consider both adversarial and stochastic regimes with heterogeneous feedback across agents. In the adversarial regime, for each round $t$ and agent $v$, the losses $\{\ell_{v,t}(k)\}_{k\in[K]}$ are fixed in advance by an adversary (before the game starts) and may differ across agents even for the same arm. In the stochastic regime, for each agent–arm pair $(v, k)$, the sequence $\{\ell_{v,t}(k)\}_{t=1}^T$ is drawn i.i.d. over time from an unknown distribution with mean $\mu_{v,k} := \mathbb{E}[\ell_{v,t}(k)]$. The performance of each agent $v$ is evaluated by its individual pseudo-regret:

$$R_T(v) = \mathbb{E}\left[\sum_{t=1}^T \overline{\ell}_t(k_{v,t})\right] - \min_{k\in[K]} \mathbb{E}\left[\sum_{t=1}^T \overline{\ell}_t(k)\right].$$

We define the globally best arm (in expectation) as $k^* \in \arg\min_{k\in[K]} \mathbb{E}\left[\sum_{t=1}^T \overline{\ell}_t(k)\right]$.

**Notations.** We denote the $K$-simplex by $\Delta^{K-1} = \{x \in \mathbb{R}_+^K \mid \|x\|_1 = 1\}$. For a convex function $F : \mathbb{R}^K \to \mathbb{R} \cup \{+\infty\}$, let $F^*$ denote its Fenchel conjugate and $\overline{F}^*$ its conjugate restricted to the simplex, i.e.,

$$F^*(y) = \max_{x\in\mathbb{R}^K}\{\langle x, y\rangle - F(x)\}, \quad \overline{F}^*(y) = \max_{x\in\Delta^{K-1}}\{\langle x, y\rangle - F(x)\}.$$

We denote by $d_v = |\mathcal{N}(v)|$ the degree of node $v$, and by $d_{\max} = \max_{v\in[V]} d_v$ the maximum node degree in the graph. For a matrix $B$, we use $\sigma_i(B)$ to denote its $i$-th largest singular value. For a real symmetric matrix $B$, we use $\lambda_i(B)$ to denote its $i$-th largest eigenvalue. The dynamics of consensus averaging among agents is typically characterized by the (unweighted) Laplacian matrix $M$ of the communication graph $G$, defined as:

$$M_{u,v} = \begin{cases} d_u & \text{if } u = v, \\ -1 & \text{if } u \neq v \text{ and } (u, v) \in E, \\ 0 & \text{otherwise.} \end{cases} \tag{1}$$

## 4 ALGORITHM

In this section, we propose our FEDFTRL method for the federated bandit problem. The details of the algorithm are presented in Algorithm 1. One challenge in federated bandits is that messages from other agents arrive with delays that depend on the network topology. This scenario can be regarded as a bandit problem with delayed feedback. Motivated by this idea, our FEDFTRL algorithm adapts the FTRL framework using a hybrid regularizer, following prior work on bandits with delayed feedback (Zimmert & Seldin, 2020; Masoudian et al., 2022). We present the regularizer used in FEDFTRL as follows:

$$F_t(x) = -2\eta_t^{-1}\left(\sum_{k=1}^K \sqrt{x_k}\right) + \gamma_t^{-1}\left(\sum_{k=1}^K x_k(\log x_k - 1)\right).$$

We introduce a time-varying parameter $C_t^P$ to quantify the delay caused by decentralized communication:

$$C_t^P = \frac{\min\{\log(Vt), \sqrt{V}\}}{1 - \sigma_2(P)} + 2 + D, \tag{2}$$

Then we set the learning-rate sequences $\{\eta_t\}_{t\geq 1}$ and $\{\gamma_t\}_{t\geq 1}$ as

$$\eta_t^{-1} = 4\sqrt{Vt + 169V^2D}, \quad \gamma_t^{-1} = 8V\sqrt{C_t^P t/\log K + 36D^2(K-1)^{\frac{2}{3}} + 4(C_t^P)^2}.$$

At each time step $t$, each agent $v$ computes a probability distribution over arms as follows:

$$x_{v,t} = \nabla \overline{F}_t^*(-\hat{L}_{v,t}^{obs}) = \arg\min_{x\in\Delta^{K-1}}\{\langle x, \hat{L}_{v,t}^{obs} + F_t(x)\rangle\}, \tag{3}$$

where $P$ is the communication matrix and $\hat{L}_{v,t}^{obs} \in \mathbb{R}^K$ is agent $v$'s cumulative loss estimator up to time $t$. The agent then samples an arm $k_{v,t} \sim x_{v,t}$ and observes a locally biased loss $\ell_{v,t}(k_{v,t})$. We construct an unbiased loss estimator and its truncated variant as follows:

$$\hat{\ell}_{v,t}(k) = \frac{\ell_{v,t}(k_{v,t})\,\mathbb{I}(k = k_{v,t})}{x_{v,t}(k)} \quad \text{and} \quad \tilde{\ell}_{v,t}(k) = \frac{\ell_{v,t}(k_{v,t})\,\mathbb{I}(k = k_{v,t})}{\max\{x_{v,t}(k), 12VC_t^P\gamma_t\}}. \tag{4}$$

---

**Algorithm 1** FEDFTRL (local routine for each agent $v$)

---

1: **Input:** a doubly stochastic matrix $P \in [0,1]^{V \times V}$; the diameter $D$ of graph $G$.
2: **Initialize:** a deviation record set $A_v \leftarrow \emptyset$; the cumulative loss estimator $\hat{L}_{v,1}^{obs} \leftarrow \mathbf{0}_K$.
3: **for** $t = 1, 2, 3, \ldots$ **do**
4:     Compute $x_{v,t} \leftarrow \arg\min_{x \in \Delta^{K-1}} \left\{ \langle x, \hat{L}_{v,t}^{obs} \rangle + F_t(x) \right\}$.
5:     Sample $k_{v,t} \sim x_{v,t}$ and observe $\ell_{v,t}(k_{v,t})$.
6:     Construct $\hat{\ell}_{v,t}$ and $\tilde{\ell}_{v,t}$ by equation 4.
7:     **if** $\hat{\ell}_{v,t}(k_{v,t}) \neq \tilde{\ell}_{v,t}(k_{v,t})$ **then**
8:         Set $w_{v,t} \leftarrow \hat{\ell}_{v,t}(k_{v,t}) - \tilde{\ell}_{v,t}(k_{v,t})$ and append $\langle v, t, k_{v,t}, w_{v,t} \rangle$ to $A_v$.
9:     **end if**
10:     Exchange the message $\{\hat{L}_{v,t}^{obs}, A_v\}$ with neighbors of agent $v$.
11:     Update cumulative loss estimator:
12:

$$\hat{L}_{v,t+1}^{obs} = \sum_{u:\, (u,v) \in E} P_{u,v} \hat{L}_{u,t}^{obs} + V \tilde{\ell}_{v,t}. \tag{5}$$

13:     Update the deviation record set: $A_v \leftarrow \bigcup_{u \in \mathcal{N}(v) \cup \{v\}} A_u$.
14:     **for** each record $\langle u, s, k, w_{u,s} \rangle \in A_v$ **do**
15:         **if** $t - s > D$ **then**
16:             Set $\hat{L}_{v,t+1}^{obs}(k) \leftarrow \hat{L}_{v,t+1}^{obs}(k) + w_{u,s}$, and remove $\langle u, s, k, w_{u,s} \rangle$ from $A_v$.
17:         **end if**
18:     **end for**
19: **end for**

---

Before communicating with neighbors at time $t$, agent $v$ prepares a message consisting of (i) its current cumulative loss estimator $\hat{L}_{v,t}^{obs}$ and (ii) a set of deviation records $A_v$. A new record $\langle v, t, k_{v,t}, w_{v,t} \rangle$ is appended to $A_v$ if and only if $\hat{\ell}_{v,t}(k_{v,t}) \neq \tilde{\ell}_{v,t}(k_{v,t})$, in which case we set the estimator's deviation as $w_{v,t} = \hat{\ell}_{v,t}(k_{v,t}) - \tilde{\ell}_{v,t}(k_{v,t})$. Next, agent $v$ averages its cumulative loss estimator with those of its neighbors and merges incoming deviation records.

### 4.1 INTUITION BEHIND THE TRUNCATED LOSS ESTIMATOR

One challenge in federated bandits is that the locally observed loss at agent $v$ can be biased in expectation relative to the average loss $\bar{\ell}_t(k)$ of the same arm. In FEDFTRL, we address this by updating $\hat{L}_{v,t+1}^{obs}$ using the *truncated* estimator $\tilde{\ell}_{v,t}$ instead of the unbiased estimator $\hat{\ell}_{v,t}$. This choice keeps all agents' action probability distributions nearly aligned.

Specifically, when constructing $\tilde{\ell}_{v,t}$ we replace the denominator with $\max\{x_{v,t}(k), 12VC_t^P \gamma_t\}$, which prevents the loss estimator from exploding when $x_{v,t}(k)$ is extremely small. As a result, no single rare arm pull can trigger an excessively large update that would cause the agents' probability distributions to diverge. This stabilization ensures well-behaved and nearly aligned action distributions across agents.

### 4.2 INTUITION BEHIND THE COMMUNICATION SCHEME

Since sharing raw observations is not allowed under the federated setting, our method communicates a deviation record set $A_v$ in each round. This communication is necessary to correct the bias introduced by the truncated estimator.

Specifically, while truncation keeps agents' action distributions nearly aligned, it can make the local loss estimates deviate from the group-average loss when truncation is active. Therefore, whenever $\hat{\ell}_{v,t}(k_{v,t}) \neq \tilde{\ell}_{v,t}(k_{v,t})$, agent $v$ appends a record $\langle v, t, k_{v,t}, w_{v,t} \rangle$ to $A_v$. Once a record $\langle u, s, k, w_{u,s} \rangle$ has propagated in the network for more than $D$ rounds (i.e., $t - s > D$), every agent will have received it. At that point, adding the correction $w_{u,s}$ to $\hat{L}_{v,t+1}^{obs}(k)$ no longer introduces distribution misalignment among agents.

Finally, recall that we scale the per-round feedback by a factor of $V$ in the cumulative loss update (see equation 5). Communication averaging drives the agents toward consensus on the group-average losses; the factor of $V$ counteracts this averaging effect, ensuring that feedback information is not overly diluted. Thus, we obtain the following regret guarantee for FEDFTRL, with the proof provided in Appendix 12.

**Theorem 1.** *If* FEDFTRL *is run with a given doubly stochastic communication matrix $P$, then in the adversarial regime, the individual regret of each agent $v$ is bounded by*

$$R_T(v) \leq 13\sqrt{KT/V} + 13\sqrt{C_T^P T \log K} + 156\sqrt{D} + 72D(K-1)^{\frac{1}{3}} \log K + 24C_T^P \log K.$$

*Furthermore, in the stochastic regime, the individual regret of each agent $v$ is bounded by*

$$R_T(v) \leq \sum_{k \neq k^*} \frac{51 \log T}{V \Delta_k} + \sum_{k \neq k^*} \frac{90 C_T^P}{\Delta_k \log K} + 56D\sqrt{(K-1)\log K} + 11C_T^P \log K.$$

*Moreover, for each agent $v$, the expected communication cost per round is $O(K)$.*

**Remark 1.** *If the doubly stochastic matrix $P$ is constructed via the* max-degree *trick (Duchi et al., 2011), i.e.,*

$$P = I - \frac{W - A}{1 + d_{\max}},$$

*where $W = \mathrm{diag}(d_1, d_2, \ldots, d_V)$ is the degree matrix and $A$ is the adjacency matrix of the communication graph $G$, then Corollary 1 of Duchi et al. (2011) implies that*

$$C_T^P = \Omega\left(\sqrt{\frac{1 + d_{\max}}{\lambda_{V-1}(M)}} \sqrt{\min\{\log(VT), \sqrt{V}\}}\right).$$

*This result shows that only low-degree polynomial gaps remain between our upper bound and the lower bound $\Omega\left(\max\{\sqrt{\frac{KT}{V}}, \sqrt[4]{\frac{1+d_{\max}}{\lambda_{V-1}(M)}} \sqrt{T \log K}\}\right)$ in the adversarial setting.*

## 5  A SKETCH OF THE PROOF OF THEOREM 1

In this section, we provide a proof sketch of Theorem 1. We present proof sketches for the regret bounds in the adversarial and stochastic settings in Section 5.1 and Section 5.2, respectively. The detailed proofs are provided in Appendix 12.

### 5.1  ADVERSARIAL BOUND

We start by stating a key lemma (Lemma 1) that controls the ratio between the action distributions of any two agents at the same time step, whose proof is provided in Section 11 in the appendix. This lemma also relates each agent's individual regret to the group regret, i.e., the sum of regrets over all agents.

**Lemma 1.** *For any two agents $u$ and $v$, and for any arm $k$ at time $t$, it holds that*

$$x_{u,t}(k) \leq \frac{3}{2} x_{v,t}(k) \quad \text{and} \quad x_{v,t}(k) \leq \frac{3}{2} x_{u,t}(k).$$

*Furthermore, for any agent $v$, its individual regret is bounded in terms of the group regret as*

$$R_T(v) \leq \frac{3}{2V} \sum_{u=1}^{V} R_T(u).$$

To bound the group regret, we transform the federated bandit into a single-agent interaction with the environment over $VT$ rounds. This reduction significantly simplifies the theoretical analysis. We introduce some additional definitions. Let the instantaneous loss estimator be

$$m_t := \frac{1}{V} \sum_{v=1}^{V} \hat{\ell}_{v,t},$$

and define the shifted cumulative loss

$$\hat{L}_{v,t} := \sum_{s=1}^{t-1} V m_s + (v-1) m_t.$$

Since we have $\mathbb{E}[m_t] = \bar{\ell}_t$, $\hat{L}_{v,t}$ can be viewed as a shifted cumulative loss with an offset proportional to $(v-1)$. This ensures that when we sum over all agents, the instantaneous loss estimator $m_t$ line up as if they were incurred sequentially by a single agent over $VT$ rounds. As a result, we can decompose the group regret into three terms:

$$\sum_{v=1}^{V} R_T(v) = \sum_{v=1}^{V} \mathbb{E}\Big[ \sum_{t=1}^{T} \big( \langle \bar{\ell}_t, x_{v,t} \rangle - \bar{\ell}_t(k^*) \big) \Big]$$

$$\leq \mathbb{E}\Big[ \underbrace{\sum_{t=1}^{T} \sum_{v=1}^{V} \Big( \bar{F}_t^*\big( -\hat{L}_{v,t}^{obs} - m_t \big) - \bar{F}_t^*\big( -\hat{L}_{v,t}^{obs} \big) + \langle x_{v,t}, m_t \rangle \Big)}_{(A)}$$

$$+ \underbrace{\sum_{t=1}^{T} \sum_{v=1}^{V} \Big( \bar{F}_t^*\big( -\hat{L}_{v,t}^{obs} \big) - \bar{F}_t^*\big( -\hat{L}_{v,t}^{obs} - m_t \big) - \bar{F}_t^*\big( -\hat{L}_{v,t} \big) + \bar{F}_t^*\big( -\hat{L}_{v+1,t} \big) \Big)}_{(B)}$$

$$+ \underbrace{\Big( \sum_{t=1}^{T} \sum_{v=1}^{V} \big( \bar{F}_t^*\big( -\hat{L}_{v,t} \big) - \bar{F}_t^*\big( -\hat{L}_{v+1,t} \big) \big) \Big)}_{(C)} - \hat{L}_{1,T+1}(k^*) \Big].$$

Term (A) is a typical Bregman-divergence term arising from the FTRL/OMD analysis, and it depends on the local norm induced by the regularizer. We can bound it as

$$\mathbb{E}[(A)] \leq \frac{9}{32} \sum_{v=1}^{V} \sum_{t=1}^{T} \sqrt{\frac{K}{Vt}} \leq \frac{9}{16} \sqrt{VKT}.$$

Term (B) is handled by the analysis in Zimmert & Seldin (2020), which yields

$$\mathbb{E}[(B)] \leq \frac{9}{32} \sum_{v=1}^{V} \sum_{t=1}^{T} \sqrt{\frac{C_t^P \log K}{t}} \leq \frac{9}{16} V \sqrt{C_T^P T \log K}.$$

Term (C) can be bounded using standard telescoping-sum techniques. In particular, we obtain

$$\mathbb{E}[(C)] \leq 8\sqrt{VKT} + 8V\sqrt{C_T^P T \log K} + 104V\sqrt{D} + 48VD(K-1)^{\frac{1}{3}} \log K + 24VC_T^P \log K.$$

Combining the bounds for (A)–(C) above and simplifying completes the proof of the adversarial bound.

## 5.2 STOCHASTIC BOUND

Inspired by the analysis of stochastic bounds for bandits with delayed feedback in Masoudian et al. (2022), let $\tilde{x}_{v,t} = \nabla \bar{F}_t^*(-\hat{L}_{v,t})$. We then define the drifted pseudo-regret as

$$R_T^{drift}(v) = \mathbb{E}\left[ \sum_{t=1}^{T} \big( \langle \tilde{x}_{v,t}, \bar{\ell}_t \rangle - \bar{\ell}_t(k^*) \big) \right].$$

We can use the drifted pseudo-regret to control the group pseudo-regret as follows:

$$\sum_{v=1}^{V} R_T(v) \leq \frac{5}{3} \sum_{v=1}^{V} R_T^{drift}(v) + VD$$

$$\leq \frac{5}{3} \underbrace{\sum_{t=1}^{T} \sum_{v=1}^{V} \sum_{k \neq k^*} \frac{5\sqrt{\tilde{x}_{v,t}(k)}}{4\sqrt{Vt}}}_{(A)} + \frac{5}{3} \underbrace{\sum_{t=2}^{T} \sum_{v=1}^{V} \sum_{k=1}^{K} \frac{C_t^P \gamma_{t-1} \tilde{x}_{v,t}(k) \log(1/\tilde{x}_{v,t}(k))}{\log K}}_{(B)}$$
$$+ \underbrace{37VD\sqrt{(K-1)\log K} + 7VC_T^P \log K + VD}_{(C)}.$$

**Self-bounding analysis:** We apply a self-bounding technique to combine terms (A) and (B) with the drifted regret. Specifically,

$$\sum_{v=1}^{V} R_T(v) \leq \frac{5}{3} \sum_{v=1}^{V} R_T^{drift}(v) + VD$$
$$\leq \frac{5}{3}\Big(3(A) - \sum_{v=1}^{V} R_T^{drift}(v) + 3(B) - \sum_{v=1}^{V} R_T^{drift}(v)\Big) + (C).$$

Using the analysis in Masoudian et al. (2022), we have

$$3(A) - \sum_{v=1}^{V} R_T^{drift}(v) \leq \sum_{k \neq k^*} \frac{81 \log T}{4\Delta_k}, \quad 3(B) - \sum_{v=1}^{V} R_T^{drift}(v) \leq \sum_{k \neq k^*} \frac{36VC_T^P}{\Delta_k \log K}.$$

Combining these bounds with (C) and simplifying yields the stated stochastic regret bound.

## 6 EXPERIMENTS

We conducted experiments on both synthetic and real-world datasets under various network topologies to evaluate the performance of FEDFTRL against several baseline methods. We consider the following baselines: FEDEXP3 (Yi & Vojnovic, 2023), GOSSIP_UCB (Zhu et al., 2021), DRBB-bandit (Zhang et al., 2025), and IND-FTRL, where IND-FTRL denotes running Tsallis-FTRL (Zimmert & Seldin, 2021) independently at each agent without communication. Following the experimental design in Yi & Vojnovic (2023), we adopt the max-degree trick to construct the doubly stochastic matrix $P$, as described in Remark 1. We set the learning-rate schedules of FEDFTRL as $\eta_t^{-1} = 0.1\sqrt{Vt}$ and $\gamma_t^{-1} = 8V\sqrt{C_t^P t / \log K}$. All experiments are repeated for 50 trials, and we report the average results as lines.

**Choice of the network graphs.** We conduct experiments on several network topologies, including a fully connected graph, a $\sqrt{V} \times \sqrt{V}$ grid graph, and a random geometric graph (RGG). An RGG-$g$ is constructed by uniformly placing each node in $[0,1]^2$ and connecting any two nodes whose Euclidean distance is at most $g$ (Penrose, 2003). In our experiments, we set $g = 0.5$.

### 6.1 SYNTHETIC DATASETS

For each agent $v$ and each arm $k$, we independently sample a mean loss $\mu_{v,k}$ uniformly from $[0,1]$. When agent $v$ pulls arm $k$ at round $t$, we draw the feedback $\ell_{v,t}(k)$ from a Gaussian distribution $\mathcal{N}(\mu_{v,k}, 0.01)$ and clip it to $[0,1]$. We set the horizon $T = 3000$, the number of agents $V = 16$, and the number of arms $K = 20$.

The results in Figure 1 show that FEDFTRL outperforms all baselines in terms of average regret. Notably, IND-FTRL cannot achieve sublinear regret when only observing locally biased feedback, highlighting the benefits of communication in our method.

### 6.2 MOVIELENS DATASET: RECOMMENDING POPULAR MOVIE GENRES

We further evaluate FEDFTRL on a real-world dataset: the MovieLens dataset (Cantador et al., 2011). This dataset contains 87,585 movies classified into 20 genres, with 32,000,204 ratings (scores in $\{0.5, 1, \ldots, 5\}$) from more than 280,000 users. Among these users, 3,963 have rated at least one

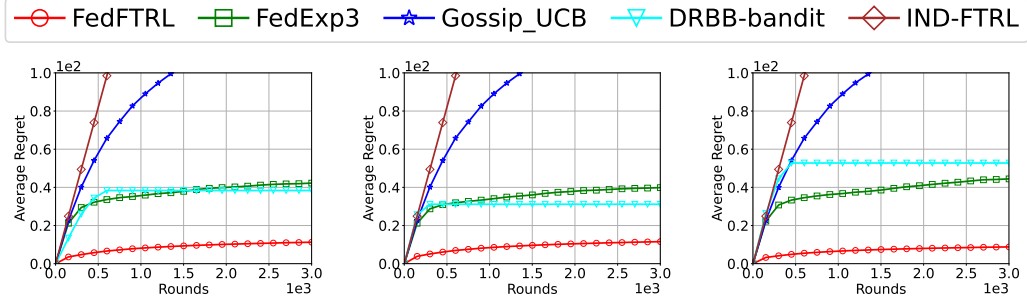

Figure 1: Average cumulative regret of FEDFTRL, FEDEXP3, IND-FTRL, GOSSIP_UCB, and DRBB-bandit on the synthetic dataset under three communication networks: (left) complete graph, (middle) grid graph, and (right) RGG-$0.5$.

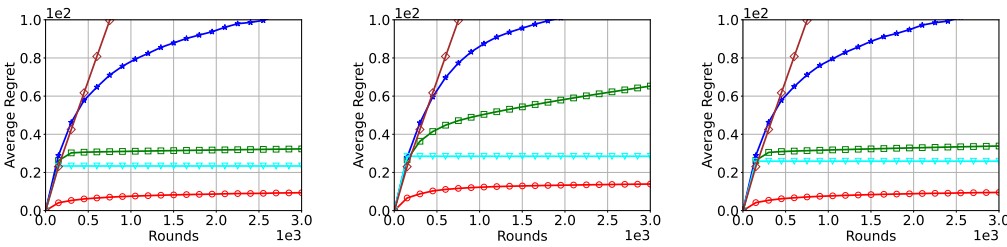

Figure 2: Average cumulative regret of FEDFTRL, FEDEXP3, IND-FTRL, GOSSIP_UCB, and DRBB-bandit on the MovieLens dataset under three communication networks: (left) complete graph, (middle) grid graph, and (right) RGG-$0.5$.

movie in every genre; we select these users as our agents and treat each genre as an arm. We set $T = 3000$, $V = 3963$, and $K = 20$.

To simulate changes in user preferences over time, we sort each user's ratings in chronological order and construct the loss sequence as follows. Let $r_v^j(k)$ be the $j$-th rating of user $v$ for genre $k$ in this sorted order, where $j \in \{1, \ldots, n_v^k\}$. The loss for user $v$ on arm $k$ at time $t$ is defined as

$$\ell_{v,t}(k) = \frac{5.5 - r_v^j(k)}{5.0} \quad \text{for} \quad t \in \left[ (j-1) \left\lfloor \frac{T}{n_v^k} \right\rfloor, j \left\lfloor \frac{T}{n_v^k} \right\rfloor \right),$$

where $n_v^k$ is the total number of ratings user $v$ has for genre $k$. In words, we partition each user's interaction timeline into $n_v^k$ segments of equal length, and assign the $j$-th rating $r_v^j(k)$ as the loss (scaled to $[0, 1]$) for all time steps in the $j$-th segment for that user–genre pair.

As shown in Figure 2, FEDFTRL significantly outperforms all baselines, demonstrating its effectiveness on real-world data.

## 6.3 SENSITIVITY TO THE TOPOLOGY PARAMETER

We study the sensitivity to the topology parameter $C_t^P$. We keep the experimental setup identical to that in Section 6 and only rescale $C_t^P$ by multiplicative factors of $0.1$, $0.5$, $1.0$ (default), $5$, and $10$. We denote the corresponding variants by FEDFTRL-$\varepsilon$, where $\varepsilon$ is the scaling factor. When $\varepsilon = 1.0$, $C_t^P$ is unchanged and we simply write FEDFTRL.

The results in Figure 3 and Figure 4 show that FEDFTRL is robust to the choice of the topology parameter $C_t^P$. Even with a misspecified $C_t^P$, our method still achieves sublinear regret.

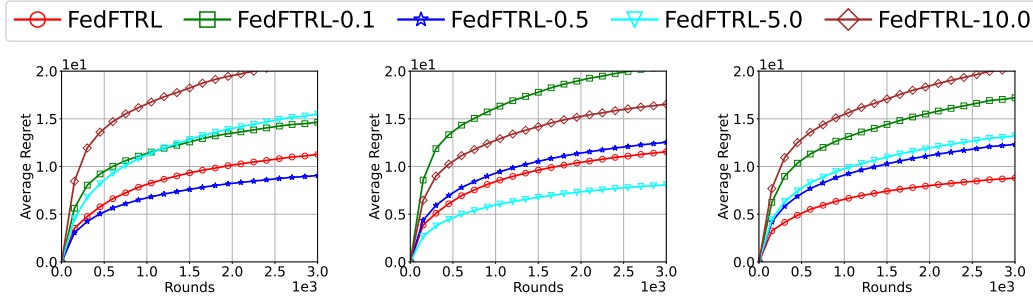

Figure 3: Average cumulative regret of FEDFTRL, FEDFTRL-0.1, FEDFTRL-0.5, FEDFTRL-5.0, and FEDFTRL-10.0 on the synthetic dataset under three communication networks: (left) complete graph, (middle) grid graph, and (right) RGG-0.5.

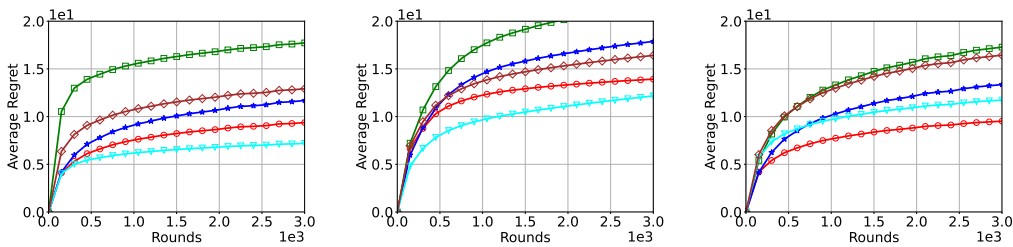

Figure 4: Average cumulative regret of FEDFTRL, FEDFTRL-0.1, FEDFTRL-0.5, FEDFTRL-5.0, and FEDFTRL-10.0 on the MovieLens dataset under three communication networks: (left) complete graph, (middle) grid graph, and (right) RGG-0.5.

## 7 CONCLUSION

In this paper, we propose a novel federated bandit algorithm, FEDFTRL, which, to the best of our knowledge, is the first to achieve a BOBW regret guarantee in both stochastic and adversarial settings. Our theoretical analysis shows that our regret upper bounds match the corresponding lower bounds up to polynomial factors. Furthermore, empirical results corroborate our theory and demonstrate the superior performance of our algorithm. An interesting future direction is to further narrow the remaining gap between the upper and lower regret bounds in the adversarial setting.

## ACKNOWLEDGMENT

The authors would like to thank the anonymous reviewers for their helpful comments. This work was supported by Natural Science Foundation of China (No. 62306116), the Key Program of National Natural Science Foundation of China (No. 62432007) and the Shanghai Special Fund for Promoting High-Quality Industrial Development (No. 2025030702).

## REPRODUCIBILITY STATEMENT

We provide complete proofs for all theoretical claims in the appendix. To facilitate reproducibility, we include the source code in the supplementary material. We use the MovieLens[1] dataset in our experiments, which is publicly available.

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

## 8 NOTATIONS

We define the instantaneous loss estimator $m_t := \frac{1}{V} \sum_{v=1}^{V} \hat{\ell}_{v,t}$, the cross-agent average cumulative loss estimator

$$\bar{L}_{t-1} := \frac{1}{V} \sum_{v=1}^{V} \hat{L}_{v,t-1}^{obs},$$

and the drifted cumulative loss

$$\hat{L}_{v,t} := \sum_{s=1}^{t-1} V m_s + (v-1) m_t.$$

For notational convenience, we adopt the wrap-around convention $(V+1, t) \equiv (1, t+1)$.

Finally, for any $x \in \Delta^{K-1}$, we introduce the *inverse-curvature weights*

$$\rho_t^k(x) \triangleq \frac{f_t''(x^k)^{-1}}{\sum_{i=1}^{K} f_t''(x^i)^{-1}}, \qquad k \in [K].$$

## 9 REGULARIZER PROPERTIES

First, we analyze some properties of the regularizer

$$F_t(x) = -2\eta_t^{-1} \sum_{k=1}^{K} x_k^{\frac{1}{2}} + \gamma_t^{-1} \sum_{k=1}^{K} x_k (\log(x_k) - 1)$$

.

Given the function $f_t(x) = -2\eta_t^{-1} \sqrt{x} + \gamma_t^{-1} x (\log(x) - 1)$.

**Fact 1.** *$f_t'(x)$ is a concave function, $f_t''(x)$ is a monotonically decreasing function, $f_t''(x)^{-1}$ is a convex function, and $f_t^{*'}$ is a convex monotonically increasing function.*

*Proof.* By definition $f_t'(x) = -\eta_t^{-1} x^{-\frac{1}{2}} + \gamma_t^{-1} \log(x)$, whose second derivative is $-\frac{3}{4}\eta_t^{-1} x^{-\frac{5}{2}} - \gamma_t^{-1} x^{-2} < 0$, which conclude the first and the second statement. $f_t''(x)^{-1} = (\frac{1}{2}\eta_t^{-1} x^{-\frac{3}{2}} + \gamma_t^{-1} x^{-1})^{-1}$ so the second derivative is

$$\frac{\eta_t \gamma_t^2 \left(2\eta_t x^{\frac{7}{2}} + 3\gamma_t x^3\right)}{2\sqrt{x} \left(2\eta_t x^{\frac{3}{2}} + \gamma_t x^{3/2}\right)^3} > 0,$$

which conclude the third claim. Since $f_t$ are Legendre functions, we have $f_t^{*''}(y) = f_t''(f_t^{*'}(y))^{-1} > 0$. Therefore the function is monotonically increasing. Since both $f_t''(x)^{-1}$, as well as $f_t^{*'}(y)$ are increasing, the composition is as well and $f_t^{*'''} > 0$. $\square$

**Fact 2.** *For any convex $F$, for $L \in \mathbb{R}^K$ and $c \in \mathbb{R}$:*

$$\bar{F}^*(L + c\mathbf{1}_K) = \bar{F}^*(L) + c.$$

*Proof.* By definition $\bar{F}^*(L + c\mathbf{1}_K) = \max_{x \in \Delta^{K-1}} \langle x, L + c\mathbf{1}_K \rangle - F(x) = \max_{x \in \Delta^{K-1}} \langle x, L \rangle - F(x) + c = \bar{F}^*(L) + c$. $\square$

**Fact 3.** *For ant $x_{v,t}$ there exists $c \in \mathbb{R}$, such that:*

$$x_{v,t} = \nabla \bar{F}_t^*(-\hat{L}_{v,t}) = \nabla F_t^*(-\hat{L}_{v,t} + c\mathbf{1}_k) = \nabla F_t^*(\nabla F_t(x_{v,t})).$$

*Proof.* By the KKT conditions, there exists $c \in \mathbb{R}$, such that $x_{v,t} = \arg\max_{x \in \Delta^{K-1}} \langle x, -\hat{L}_{v,t} \rangle + F_t(x)$ satisfies $\nabla F_t(x_{v,t}) = -\hat{L}_{v,t} + c\mathbf{1}_K$. The rest follows by the standard property $\nabla F = (\nabla F^*)^{-1}$ of Legendre $F$. $\square$

**Fact 4.** *For any Legendre function $F$ and $L \in \mathbb{R}^K$ it holds that*

$$\bar{F}^*(L) \leq F^*(L),$$

*with equality if and only if there exists $x \in \Delta^{K-1}$ such that $L = \nabla F(x)$.*

*Proof.* The first statement follows from the definition, since for any $A \subset B$:$\max_{x \in A} f(x) \leq \max_{x \in B} f(x)$. The second part follows because equality means that $\arg\max_x (\langle x, L \rangle - F(x)) = \nabla F^*(L) \in \Delta^{K-1}$, which is equivalent to the statement. $\square$

**Fact 5.** *For any $x \in \Delta^{K-1}$, $L \geq 0$ and $k \in [K]$, we have*

$$\left(\nabla \bar{F}_t^*(\nabla F_t(x) - L)\right)_k \geq \left(\nabla F_t^*(\nabla F_t(x) - L)\right)_k.$$

*Proof.* By Fact 3, there exists some $c \in \mathbb{R}$ such that $\nabla \bar{F}_t^*(\nabla F_t(x) - L) = \nabla F_t^*(\nabla F_t(x) - L + c\mathbf{1}_K)$. The statement is equivalent to $c$ being non-negative, since $F_t^{*'}$ are monotonically increasing. If $c < 0$, then

$$1 = \sum_{k=1}^{K} \left(\nabla \bar{F}_t^*(\nabla F_t(x) - L)\right)_k$$
$$= \sum_{k=1}^{K} \left(\nabla F_t^*(\nabla F_t(x) - L + c\mathbf{1}_K)\right)_k = \sum_{k=1}^{K} F_t^{*'}\left(F_t'(x_k) - L_k + c\right) < \sum_{k=1}^{K} F_t^{*'}\left(F_t'(x_k)\right) = 1,$$

which is a contradiction. Hence $c$ must be non-negative, and the proof is complete. $\square$

**Fact 6.** *Let $D_F(x, y) = F(x) - F(y) - \langle x - y, \nabla F(y) \rangle$ be the Bregman divergence of a function $F$. For any Legendre function $f$ with monotonically decreasing second derivative, $x \in \mathrm{dom}(f)$, and $\ell \geq 0$, such that $f'(x) - \ell \in \mathrm{dom}(f^*)$, we have*

$$D_{f^*}\left(f'(x) - \ell, f'(x)\right) \leq \frac{\ell^2}{2 f''(x)}.$$

*Proof.* By Taylor's theorem, there exists some $\tilde{x} \in \left[f^{*'}(f'(x) - \ell), x\right]$ such that

$$D_{f^*}\left(f'(x) - \ell, f'(x)\right) = \frac{\ell^2}{2 f''(\tilde{x})}.$$

Note that $\tilde{x}$ is smaller than $x$, since $f^{*'}$ is monotonically increasing. Finally, using the fact that the second derivative is decreasing allows us to bound

$$f''(\tilde{x})^{-1} \leq f''(x)^{-1}.$$

Hence the stated inequality follows. $\square$

**Fact 7.** *For any convex function $F$, and $L_2 \geq L_1$ (coordinate wise), we have*

$$\bar{F}^*(-L_1) \geq \bar{F}^*(-L_2).$$

*Proof.* We have

$$\begin{aligned}
\bar{F}^*(-L_2) &= \left\langle \nabla \bar{F}^*(-L_2), -L_2 \right\rangle + F\left(\nabla \bar{F}^*(-L_2)\right) \\
&\leq \left\langle \nabla \bar{F}^*(-L_2), -L_1 \right\rangle + F\left(\nabla \bar{F}^*(-L_2)\right) \\
&\leq \max_{x \in \Delta^{K-1}} \left(\langle x, -L_1 \rangle + F(x)\right) \\
&= \bar{F}^*(-L_1).
\end{aligned}$$

$\square$

## 10 AUXILIARY LEMMAS

In this section, we present several preliminary lemmas that will be used in the subsequent proofs.

**Lemma 2.** *For any time step $t$ and agent $v \in [V]$, we have*

$$\|\bar{L}_t - \hat{L}_{v,t}^{obs}\|_\infty = \|\hat{L}_{v,t}^{obs} - \bar{L}_t\|_\infty \leq \frac{1}{12\gamma_t}.$$

*Proof.* As mentioned before, updating with deviation records does not affect $\|\bar{L}_t - \hat{L}_{v,t}^{obs}\|_\infty$, since all agents perform this operation. Hence, we only consider the impact of equation 5.

We first record a simple bound that holds for all $v, t$:

$$\|\tilde{\ell}_{v,t}\|_\infty = \frac{\ell_{v,t}(k_{v,t})}{\max\{x_{v,t}(k_{v,t}), 12VC_t^P\gamma_t\}} \leq \frac{1}{12VC_t^P\gamma_t}.$$

Since $\{\gamma_t\}$ is non-increasing, applying Lemma 6 in Hosseini et al. (2013) with $L = \frac{1}{12C_t^P\gamma_t}$ yields

$$\|\bar{L}_t - \hat{L}_{v,t}^{obs}\|_\infty \leq \frac{1}{12C_t^P\gamma_t}\Big(\frac{\sqrt{V}}{1-\sigma_2(P)} + 2\Big). \tag{6}$$

Following the definition of $\bar{L}_t$, we have

$$
\begin{aligned}
\|\bar{L}_t - \hat{L}_{v,t}^{obs}\|_\infty &= V\Big\|\sum_{s=1}^{t-1}\sum_{u=1}^{V}\Big(\mathbf{1}_K/V - P_{u,v}^{t-s+1}\Big)\tilde{\ell}_{u,s} + \Big(\frac{1}{V}\sum_{u=1}^{V}\tilde{\ell}_{u,t} - \tilde{\ell}_{v,t}\Big)\Big\|_\infty \\
&\leq V\sum_{s=1}^{t-1}\sum_{u=1}^{V}\|\tilde{\ell}_{u,s}\|_\infty \big\|\mathbf{1}_K/V - P_{u,v}^{t-s+1}\big\|_1 + \sum_{u=1}^{V}\|\tilde{\ell}_{u,t} - \tilde{\ell}_{v,t}\|_\infty \\
&\leq \sum_{s=1}^{t-1}\frac{1}{12C_t^P\gamma_t}\big\|P_{u,v}^{t-s+1} - \mathbf{1}_K/V\big\|_1 + \frac{1}{12C_t^P\gamma_t}. \tag{7}
\end{aligned}
$$

From (23) in Duchi et al. (2011), $\|P_{u,v}^{t-s+1} - \mathbf{1}_K/V\|_1 \leq \sqrt{V}\,\sigma_2(P)^{t-s+1}$. Hence, if

$$t - s \geq \frac{\log(\epsilon^{-1})}{\log(\sigma_2(P)^{-1})} - 1, \quad \text{then} \quad \|P_{u,v}^{t-s+1} - \mathbf{1}_K/V\|_1 \leq \sqrt{V}\,\epsilon.$$

Setting $\epsilon^{-1} = Vt$, for $t - s + 1 \geq \frac{\log(Vt)}{\log(\sigma_2(P)^{-1})}$ we obtain

$$\|P_{u,v}^{t-s+1} - \mathbf{1}_K/V\|_1 \leq \frac{1}{t}. \tag{8}$$

For the remaining terms, we simply use $\|P_{u,v}^{t-s+1} - \mathbf{1}_K/V\|_1 \leq 1$. The above suggests splitting the sum at $\hat{t} = \frac{\log(Vt)}{\log(\sigma_2(P)^{-1})}$. Using equation 7 and equation 8, we obtain

$$\|\bar{L}_t - \hat{L}_{v,t}^{obs}\|_\infty \leq \frac{1}{12C_t^P\gamma_t}\left(\frac{\log(Vt)}{\log(\sigma_2(P)^{-1})} + 2\right) \leq \frac{1}{12\gamma_t}. \tag{9}$$

The last inequality follows since $\log(\sigma_2(P)^{-1}) \geq 1 - \sigma_2(P)$. Combining equation 6 and equation 9 completes the proof. $\qquad\square$

**Lemma 3.** *Fix $t$ and let $x_1 = \nabla\bar{F}_t^*(-L_1)$ and $x_2 = \nabla\bar{F}_t^*(-L_2)$. For any arm $k \in [K]$, assume that*

$$\sum_{i=1}^{K}\rho_t^i(x_1)\big(L_2(i) - L_1(i)\big) \leq \frac{\alpha_1}{\gamma_t}, \quad L_1(k) - L_2(k) \leq \frac{\alpha_2}{\gamma_t},$$

*where $\alpha_1, \alpha_2 \in \big[0, \frac{1}{2}\big)$. Then*

$$x_2^k \leq \frac{1}{1 - \alpha_1 - \alpha_2}\,x_1^k.$$

*Proof.* Let $\mu_1, \mu_2$ be the KKT multipliers associated with the simplex constraint, so that for all $i \in [K]$,

$$f_t'(x_1^i) = -L_1(i) + \mu_1, \qquad f_t'(x_2^i) = -L_2(i) + \mu_2.$$

Since $f_t'$ is concave (Fact 1), for every $i \in [K]$ we have

$$(x_1^i - x_2^i) f_t''(x_1^i) \le f_t'(x_1^i) - f_t'(x_2^i) \le (x_1^i - x_2^i) f_t''(x_2^i). \tag{10}$$

Using the left inequality in equation 10 and $f_t''(x_1^i) \ge 0$ yields

$$x_1^i - x_2^i \le f_t''(x_1^i)^{-1}(\mu_1 - \mu_2 + L_2(i) - L_1(i)).$$

Summing over $i$ and using $\sum_{i=1}^K x_1^i = \sum_{i=1}^K x_2^i = 1$ gives

$$0 = \sum_{i=1}^K (x_1^i - x_2^i) \le \sum_{i=1}^K f_t''(x_1^i)^{-1}(\mu_1 - \mu_2 + L_2(i) - L_1(i)),$$

and hence

$$\mu_2 - \mu_1 \le \frac{\sum_{i=1}^K f_t''(x_1^i)^{-1}(L_2(i) - L_1(i))}{\sum_{j=1}^K f_t''(x_1^j)^{-1}} = \sum_{i=1}^K \rho_t^i(x_1)(L_2(i) - L_1(i)) \le \frac{\alpha_1}{\gamma_t}. \tag{11}$$

Next, applying the right inequality in equation 10 with $i = k$ and using $f_t'(x_2^k) - f_t'(x_1^k) = \mu_2 - \mu_1 + L_1(k) - L_2(k)$, we obtain

$$(x_2^k - x_1^k) f_t''(x_2^k) \le \mu_2 - \mu_1 + L_1(k) - L_2(k).$$

Therefore, by equation 11 and the assumption $L_1(k) - L_2(k) \le \alpha_2/\gamma_t$,

$$x_2^k \le x_1^k + f_t''(x_2^k)^{-1}(\mu_2 - \mu_1 + L_1(k) - L_2(k))$$
$$\le x_1^k + f_t''(x_2^k)^{-1}\left(\frac{\alpha_1 + \alpha_2}{\gamma_t}\right). \tag{12}$$

Finally, using the structural bound $f_t''(x)^{-1} \le \gamma_t x$ (which holds for our choice of $f_t$), we have $\frac{f_t''(x_2^k)^{-1}}{\gamma_t} \le x_2^k$, and thus

$$x_2^k \le x_1^k + (\alpha_1 + \alpha_2) x_2^k,$$

which rearranges to $x_2^k \le \frac{1}{1-\alpha_1-\alpha_2} x_1^k$. □

## 11 PROOF OF LEMMA 1

*Proof.* By Lemma 2, for any agents $u, v \in [V]$, we have

$$\|\hat{L}_{v,t}^{obs} - \hat{L}_{u,t}^{obs}\|_\infty \le \|\hat{L}_{v,t}^{obs} - \bar{L}_t\|_\infty + \|\bar{L}_t - \hat{L}_{u,t}^{obs}\|_\infty \le \frac{1}{6\gamma_t}.$$

Consequently,

$$\sum_{i=1}^K \rho_t^i(x_{u,t})(\hat{L}_{v,t}^{obs}(i) - \hat{L}_{u,t}^{obs}(i)) \le \sum_{i=1}^K \rho_t^i(x_{u,t}) \|\hat{L}_{v,t}^{obs} - \hat{L}_{u,t}^{obs}\|_\infty = \|\hat{L}_{v,t}^{obs} - \hat{L}_{u,t}^{obs}\|_\infty \le \frac{1}{6\gamma_t}.$$

Moreover, for any $k \in [K]$,

$$\hat{L}_{u,t}^{obs}(k) - \hat{L}_{v,t}^{obs}(k) \le \|\hat{L}_{u,t}^{obs} - \hat{L}_{v,t}^{obs}\|_\infty = \le \|\hat{L}_{v,t}^{obs} - \hat{L}_{u,t}^{obs}\|_\infty \le \frac{1}{6\gamma_t}.$$

Applying Lemma 3 with $\alpha_1 = \alpha_2 = \frac{1}{6}$ yields

$$x_{v,t}^k \le \frac{1}{1 - \frac{1}{6} - \frac{1}{6}} x_{u,t}^k = \frac{3}{2} x_{u,t}^k.$$

Swapping the roles of $u$ and $v$ gives the reverse inequality

$$x_{u,t}^k \le \frac{3}{2} x_{v,t}^k,$$

which completes the proof. □

## 12 PROOF OF THEOREM 1

### 12.1 LEMMAS

**Lemma 4.** *Fix any agents $u, v \in [V]$ and time indices $s, t \in [T]$. Suppose there exists a constant $\alpha \geq 1$ such that $x_{v,t}(k) \leq \alpha \, x_{u,s}(k)$ for all $k \in [K]$. Then,*

$$\sum_{k=1}^{K} \rho_t^k(x_{v,t})\tilde{\ell}_{u,s}(k) \leq \sum_{k=1}^{K} \rho_t^k(x_{v,t})\hat{\ell}_{u,s}(k) \leq 2\alpha(K-1)^{\frac{1}{3}}.$$

*Proof.* Now we aim to bound:

$$
\begin{aligned}
\sum_{k=1}^{K} \rho_t^k(x_{v,t})\hat{\ell}_{u,s}(k) &= \frac{f''(x_{v,t}(k_{u,s}))^{-1}x_{u,s}(k_{u,s})^{-1}\ell_{u,s}(k_{u,s})}{\sum_{k=1}^{K} f''(x_{v,t}^k)^{-1}} \\
&\leq \frac{f''(x_{v,t}(k_{u,s}))^{-1}x_{v,t}(k_{u,s})^{-1}(x_{v,t}(k_{u,s})/x_{u,s}(k_{u,s}))}{\sum_{k=1}^{K} f''(x_{v,t}^k)^{-1}} \\
&\leq \frac{f''(x_{v,t}(k_{u,s}))^{-1}\alpha x_{v,t}(k_{u,s})^{-1}}{\sum_{k=1}^{K} f''(x_{v,t}^k)^{-1}} \\
&\leq \frac{\alpha f''(x_{v,t}(k_{u,s}))^{-1}x_{v,t}(k_{u,s})^{-1}}{(K-1)f''\left(\frac{1-x_{v,t}(k_{u,s})}{K-1}\right)^{-1} + f''(x_{v,t}(k_{u,s}))^{-1}} \quad \text{Define } z := x_{v,t}(k_{u,s}) \\
&= \frac{\alpha(\eta_t^{-1}z^{-3/2} + 2\gamma_t^{-1}z^{-1})^{-1}z^{-1}}{(K-1)(\eta_t^{-1}(\frac{1-z}{K-1})^{-3/2} + 2\gamma_t^{-1}(\frac{1-z}{K-1})^{-1})^{-1} + (\eta_t z^{-3/2} + 2\gamma_t^{-1}z^{-1})^{-1}} \\
&= \alpha\left((1-z)\frac{\eta_t^{-1}z^{-1/2} + 2\gamma_t^{-1}}{\eta_t^{-1}\sqrt{K-1}(1-z)^{-1/2} + 2\gamma_t^{-1}} + z\right)^{-1} \quad (13)
\end{aligned}
$$

where the first inequality follows by $\ell_{u,s}(k_{u,s}) \leq 1$, and the third inequality is due to convexity of $f''(x)^{-1}$ from Fact 1. Now for $z$ we have two cases, $z < \frac{1}{K}$ and $z \geq \frac{1}{K}$.

a) $z \leq \frac{1}{K}$: This case implies

$$\frac{1-z}{z} = \frac{1}{z} - 1 \geq K - 1 \Rightarrow (1-z)^{-1/2}\sqrt{K-1} \leq z^{-1/2}$$

$$\Rightarrow 1 \leq \frac{\eta_t^{-1}z^{-1/2} + 2\gamma_t^{-1}}{\eta_t^{-1}\sqrt{K-1}(1-z)^{-1/2} + 2\gamma_t^{-1}} \quad (14)$$

Plugging equation 14 into equation 13 gives us

$$\frac{\sum_{k=1}^{K} f''(x_{v,t}^k)^{-1}\hat{\ell}_{u,s}(k)}{\sum_{k=1}^{K} f''(x_{v,t}^k)^{-1}} \leq \alpha(1-z+z)^{-1} = \alpha$$

b) $z \geq \frac{1}{K}$: Similar to previous case $z \geq \frac{1}{K}$ implies $\eta_t^{-1}z^{-1/2} \leq \eta_t^{-1}\sqrt{K-1}(1-z)^{-1/2}$ so the minimum of $\frac{\eta_t^{-1}z^{-1/2}+2\gamma_t^{-1}}{\eta_t^{-1}\sqrt{K-1}(1-z)^{-1/2}+2\gamma_t^{-1}}$ occurs when $2\gamma_t^{-1} = 0$. So substituting $2\gamma_t^{-1} = 0$ in equation 13 leads us to have

$$\frac{\sum_{k=1}^{K} f''(x_{v,t}^k)^{-1}\hat{\ell}_{u,s}(k)}{\sum_{k=1}^{K} f''(x_{v,t}^k)^{-1}} \leq \alpha^{-1}((1-z)^{3/2}z^{-1/2}(K-1)^{-1/2} + z)^{-1} \quad (15)$$

In this case again we have two following cases

b1) $z \geq \frac{1}{(K-1)^{1/3}+1}$: With this we have

$$\alpha((1-z)^{3/2}z^{-1/2}(K-1)^{-1/2}+z)^{-1} \leq \alpha z^{-1} \leq \alpha V\left((K-1)^{1/3}+1\right) \leq 2\alpha(K-1)^{1/3}$$

b2) $z \leq \frac{1}{(K-1)^{1/3}+1}$: This tells us that $(1-z) \geq \frac{(K-1)^{1/3}}{(K-1)^{1/3}+1} \geq \frac{1}{2}$ where we can use it in equation 15 as the following

$$\alpha \left( (1-z)^{3/2} z^{-1/2} (K-1)^{-1/2} + z \right)^{-1}$$

$$\leq \alpha \left( \frac{z^{-1/2}(K-1)^{-1/2}}{\sqrt{8}} + z \right)^{-1}$$

$$= \alpha \left( \frac{z^{-1/2}(K-1)^{-1/2}}{2\sqrt{8}} + \frac{z^{-1/2}(K-1)^{-1/2}}{2\sqrt{8}} + z \right)^{-1}$$

$$\leq \frac{\alpha}{3} \left( \frac{(K-1)^{-1}}{32} \right)^{-1/3} \leq 2\alpha (K-1)^{1/3}$$

where the second inequality uses AM-GM inequality.

So at the end combining results of all cases to complete the proof. $\qquad\square$

**Lemma 5.** *Fix any time indices $s, t \in [T]$ and a loss vector $L \in \mathbb{R}^K$. Let*

$$x_1 = \nabla \bar{F}_s^*(-L), \qquad x_2 = \nabla \bar{F}_t^*(-L).$$

*If $s \leq t - D$, then for every arm $k \in [K]$,*

$$x_2^k \leq \frac{5}{4} x_1^k.$$

*Proof.* Since $x_1 = \nabla \bar{F}_s^*(-\hat{L})$ and $x_2 = \nabla \bar{F}_t^*(-\hat{L})$, by the KKT conditions $\exists \mu_1, \mu_2$ s.t. $\forall k$:

$$f_s^{'}(x_1^k) = -L(k) + \mu_1, \quad f_t^{'}(x_2^k) = -L(k) + \mu_2.$$

We also know that $\exists\, k : x_1^k \geq x_2^k$ which leads to have

$$-L(k) + \mu_2 = f_t^{'}(x_2^k) \leq f_s^{'}(x_2^k) \leq f_s^{'}(x_1^k) = -L_{(}k) + \mu_1,$$

where the first inequality holds because the learning rates are decreasing and the second inequality is due to the fact that $f_s^{'}(x)$ is increasing. This implies that $\mu_2 \leq \mu_1$ which gives us the following inequality for all $k$:

$$f_t^{'}(x_2^k) = -\frac{1}{\eta_t \sqrt{x_2^k}} + \gamma_t^{-1} \log(x_2^k) \leq -\frac{1}{\eta_s \sqrt{x_1^k}} + \gamma_s^{-1} \log(x_1^k) = f_s^{'}(x_1^k).$$

Define $\beta = x_2^k / x_1^k$. So using above inequality we have

$$\frac{1}{\eta_s \sqrt{x_1^k}} - \gamma_s^{-1} \log(x_1^k) \leq \frac{1}{\eta_t \sqrt{\beta x_1^k}} - \gamma_t^{-1} \log(x_1^k) - \gamma_t^{-1} \log(\beta)$$

$$\Rightarrow \frac{1}{\sqrt{\beta}} \geq \frac{\eta_t}{\eta_s} + 2\sqrt{x_1^k} \log(\sqrt{x_1^k}) \left( \frac{\eta_t}{\gamma_t} - \frac{\eta_t}{\gamma_s} \right) + \log(\beta) \frac{\eta_t}{\gamma_t} \sqrt{x_1^k}$$

$$\geq \frac{\eta_t}{\eta_s} + \min_{0 < z \leq 1} \left\{ 2z \log(z) \left( \frac{\eta_t}{\gamma_t} - \frac{\eta_t}{\gamma_s} \right) + \log(\beta) \frac{\eta_t}{\gamma_t} z \right\}$$

$$\overset{(a)}{=} \frac{\eta_t}{\eta_s} - \frac{2}{e} \left( \frac{\eta_t}{\gamma_t} - \frac{\eta_t}{\gamma_s} \right) \left( \frac{1}{\sqrt{\beta}} \right)^{\frac{\gamma_t^{-1}}{\gamma_t^{-1} - \gamma_s^{-1}}}$$

$$\overset{(b)}{\geq} \frac{\eta_t}{\eta_s} - \frac{2}{e} \left( \frac{\eta_t}{\gamma_t} - \frac{\eta_t}{\gamma_s} \right) \frac{1}{\sqrt{\beta}}.$$

$\qquad\square$

where (a) holds because the subject function of the minimization problem is convex and equating the first derivative to zero gives $z = \beta^{\frac{\gamma_t^{-1}}{\gamma_t^{-1} - \gamma_s^{-1}}}$, and (b) follows by $\frac{\gamma_t^{-1}}{\gamma_t^{-1} - \gamma_s^{-1}} \geq 1$. So rearranging the above result gives

$$\beta \leq \left( \frac{\eta_s}{\eta_t} + \frac{2}{e} \left( \frac{\eta_t}{\gamma_t} - \frac{\eta_t}{\gamma_s} \right) \right)^2. \tag{16}$$

Therefore, we have

$$\frac{\eta_s}{\eta_t} = \frac{4\sqrt{Vt + 169V^2D}}{4\sqrt{Vs + 169V^2D}} = \sqrt{1 + \frac{V(t-s)}{169V^2D}} \leq \sqrt{1 + \frac{1}{169}}.$$

where $V \geq 1$, $D \geq 1$ and $t - s \leq D$. First, we give an inequality

$$\sqrt{x+a} - \sqrt{y+a} \leq \sqrt{x-y}, \quad x \geq y \geq 0, \quad a \geq 0.$$

We square the left side:

$$(\sqrt{x+a} - \sqrt{y+a})^2 = x+a-2\sqrt{x+a}\sqrt{y+a}+y+a \leq x+a-2\sqrt{y+a}\sqrt{y+a}+y+a = x-y.$$

By this inequality, we have

$$\frac{2}{e} \left( \frac{\eta_t}{\gamma_t} - \frac{\eta_t}{\gamma_s} \right) \leq \frac{2}{e} \left( \frac{2V(\sqrt{t-s})}{\sqrt{Vt + 169V^2D}} \right) \leq \frac{4}{e} \left( \frac{V\sqrt{t-s}}{13V\sqrt{D}} \right) \leq \frac{4}{13e}.$$

Plugging the above inequalities gives us the following bound:

$$\beta \leq \left( \sqrt{1 + \frac{1}{169}} + \frac{4}{13e} \right)^2 < \frac{5}{4}.$$

**Lemma 6.** *For any time index $s \leq t - D$ and any fixed arm $k \in [K]$, it holds that*

$$x_{v,t}^k \leq 2x_{u,s}^k.$$

*Proof.* First, we decompose $\hat{L}_{v,t}^{obs}$ into the following two parts:

$$\hat{L}_{v,t}^{obs} = \hat{L}_{v,1 \to s}^{obs} + \hat{L}_{v,s+1 \to t}^{obs},$$

where the former represents the cumulative loss estimate observed by agent $v$ from time step 1 to $s$, and the latter is the cumulative loss estimate from time step $s + 1$ to $t$.

Using the same analytical method in Lemma 2, we can obtain:

$$\|\hat{L}_{u,s}^{obs} - \hat{L}_{v,t}^{obs}\|_\infty \leq \|\hat{L}_{u,s}^{obs} - \hat{L}_{v,1 \to s}^{obs}\|_\infty \leq \|\hat{L}_{u,s}^{obs} - \bar{L}_s\|_\infty + \|\bar{L}_s - \hat{L}_{v,1 \to s}^{obs}\|_\infty \leq \frac{1}{6\gamma_t}. \tag{17}$$

Where the first inequality because that $\hat{L}_{v,s+1 \to t}^{obs}(k) \geq 0$.

Using Lemma 2, for any fixed $k$ we have

$$\hat{L}_{v,1 \to s}^{obs}(k) - \hat{L}_{u,s}^{obs}(k) \leq V \sum_{\hat{t}=s-D}^{t-D} m_{\hat{t}}(k) + \|\hat{L}_{v,1 \to s}^{obs} - L_{u,s}^{obs}(k)\|_\infty$$

$$\leq V \sum_{\hat{t}=s-D}^{t-D} m_{\hat{t}}(k) + \frac{1}{12C_t^P \gamma_t} \left( \frac{\min\{\sqrt{V}, \log(Vt)\}}{1 - \sigma_2(P)} + 2 \right). \tag{18}$$

Where the first inequality because only the deviation generated in time period $[s - D, t - D]$ will be used in $\hat{L}_{v,1 \to s}(k)$.

By Lemma 4 and mathematical induction, we have

$$\sum_{\hat{t}=s-D}^{t-D} \sum_{i=1}^{V} \sum_{k=1}^{K} \rho_t^k(x_{v,t}) m_{\hat{t}}(k) \leq \sum_{\hat{t}=s-D}^{t-D} \sum_{i=1}^{V} \sum_{k=1}^{K} \rho_t^k(x_{v,t}) \hat{\ell}_{i,\hat{t}}(k) \leq 4VD(K-1)^{\frac{1}{3}}. \tag{19}$$

Furthermore, we have

$$\hat{L}^{obs}_{v,s+1\to t}(k) = V \sum_{\hat{t}=s+1}^{t-1} \sum_{u=1}^{V} P^{t-\hat{t}-1}_{u,v} \tilde{\ell}_{u,t}(k) + \tilde{\ell}_{v,t}(k) \leq \frac{D}{12C^P_t \gamma_t}. \tag{20}$$

Combine equation 18, equation 19 and equation 20, we can complete the right statement

$$\sum_{k=1}^{K} \rho^k_t(x_{u,s})(\hat{L}^{obs}_{v,t}(k) - \hat{L}^{obs}_{u,s}(k))$$

$$= \sum_{k=1}^{K} \rho^k_t(x_{u,s})(\hat{L}^{obs}_{v,1\to s}(k) - \hat{L}^{obs}_{u,s}(k)) + \sum_{k=1}^{K} \rho^k_t(x_{u,s})\hat{L}^{obs}_{v,s+1\to t}(k)$$

$$\leq \frac{1}{12C^P_t \gamma_t}\left(\frac{\min\{\sqrt{V}, \log(Vt)\}}{1 - \sigma_2(P)} + 2 + D\right) + 4VD(K-1)^{\frac{1}{3}}$$

$$\leq \frac{1}{6\gamma_t}.$$

Where the last inequality uses the fact

$$\gamma_t^{-1} = 8V\sqrt{C^P_t t / \log K + 36D^2(K-1)^{2/3} + 4(C^P_t)^2} \geq 48VD(K-1)^{\frac{1}{3}}.$$

Using Lemma 3 and Lemma 5, we can complete the proof:

$$x^k_{v,t} \leq \frac{1}{1 - \frac{1}{6} - \frac{1}{6}} \times \frac{5}{4}x^k_{u,s} \leq 2x^k_{u,s}.$$

$\square$

**Lemma 7.** *For any time index $s = t - D$, any agents $u, v \in [V]$ and any fixed arm $k \in [K]$. Define*

$$\tilde{x}_{u,s} \triangleq \nabla \bar{F}^*_t(-\hat{L}_{u,s})$$

*Then, for every arm $k \in [K]$,*

$$x^k_{v,t} \leq \frac{5}{3}\tilde{x}^k_{u,s},$$

*Proof.* First, we decompose $\hat{L}^{obs}_{v,t}$ into the following two parts:

$$\hat{L}^{obs}_{v,t} = \hat{L}^{obs}_{v,1\to s} + \hat{L}^{obs}_{v,s+1\to t},$$

where the former represents the cumulative loss estimate observed by agent $v$ from time step 1 to $s$, and the latter is the cumulative loss estimate from time step $s + 1$ to $t$.

For any arm $k \in [K]$, using the same analytical method in Lemma 2, we can obtain:

$$\hat{L}_{u,s}(k) - \hat{L}^{obs}_{v,t}(k) < \hat{L}_{1,s+1}(k) - \hat{L}^{obs}_{v,1\to s}(k) \leq \bar{L}_s(k) - \hat{L}^{obs}_{v,s}(k) \leq \frac{1}{12\gamma_t}. \tag{21}$$

The first inequality uses the fact that $\hat{L}^{obs}_{v,s+1\to t}(k) \geq 0$ for all $k \in [K]$.

Using Lemma 2, for any fixed $k$ we have

$$\hat{L}^{obs}_{v,1\to s}(k) - \hat{L}_{u,s}(k) \leq (V-u)m_s(k) + \hat{L}^{obs}_{v,1\to s}(k) - \hat{L}_{1,s+1}(k)$$

$$\leq Vm_s(k) + \frac{1}{12C^P_t \gamma_t}\left(\frac{\min\{\sqrt{V}, \log(Vt)\}}{1 - \sigma_2(P)} + 2\right). \tag{22}$$

By Lemma 4, we have

$$\sum_{k=1}^{K} \rho^k_t(x_{v,t})Vm_s(k) = \sum_{k=1}^{K} \sum_{i=1}^{V} \rho^k_t(x_{v,t})\hat{\ell}_{i,s}(k) \leq 4V(K-1)^{\frac{1}{3}}. \tag{23}$$

According to our update rules for deviation records, no records generated in time period $[t-D+1, t]$ will be used, so we have

$$\hat{L}^{obs}_{v,s+1\to t}(k) = V \sum_{\hat{t}=s+1}^{t-1} \sum_{u=1}^{V} P^{t-\hat{t}-1}_{u,v} \tilde{\ell}_{u,t}(k) + \tilde{\ell}_{v,t}(k) \le \frac{D}{12C^P_t \gamma_t}. \tag{24}$$

Combine equation 22, equation 23 and equation 24, we have

$$\sum_{k=1}^{K} \rho^k_t(\tilde{x}_{u,s})(\hat{L}^{obs}_{v,t}(k) - \hat{L}_{u,s}(k))$$

$$= \sum_{k=1}^{K} \rho^k_t(\tilde{x}_{u,s})(\hat{L}^{obs}_{v,1\to s}(k) - \hat{L}_{u,s}(k)) + \sum_{k=1}^{K} \rho^k_t(\tilde{x}_{u,s})\hat{L}^{obs}_{v,s+1\to t}(k)$$

$$\le \frac{1}{12C^P_t \gamma_t} \left( \frac{\min\{\sqrt{V}, \log(Vt)\}}{1 - \sigma_2(P)} + 2 + D \right) + 4V(K-1)^{\frac{1}{3}}$$

$$\le \frac{1}{6\gamma_t}.$$

Where the last inequality uses the fact

$$\gamma_t^{-1} = 8V\sqrt{C^P_t t / \log K + 36D^2(K-1)^{2/3} + 4(C^P_t)^2} \ge 48V(K-1)^{\frac{1}{3}}.$$

Using Lemma 3 and Lemma 5, we can complete the proof:

$$x^k_{v,t} \le \frac{1}{1 - \frac{1}{12} - \frac{1}{6}} \times \frac{5}{4}\tilde{x}^k_{u,s} \le \frac{5}{3}\tilde{x}^k_{u,s}.$$

$\square$

**Lemma 8.** *For any round $t \in [T]$, any arm $k \in [K]$, and any two agents $u, v \in [V]$, define*

$$\tilde{x}^k_{u,t} = \nabla \bar{F}^*_t(-\hat{L}_{u,t})$$

*Then*

$$\tilde{x}^k_{u,t} \le 2x^k_{v,t}.$$

*Proof.* Using the same analytical method in Lemma 2, we can obtain:

$$\|\hat{L}^{obs}_{v,t} - \hat{L}_{u,t}\|_\infty \le \|\hat{L}^{obs}_{v,t-1} - \bar{L}_{t-1}\|_\infty + \|\tilde{\ell}_{v,t}\|_\infty$$

$$\le \frac{1}{12C^P_t \gamma_t} \left( \frac{\min\{\sqrt{V}, \log(Vt)\}}{1 - \sigma_2(P)} + 2 \right) + \frac{1}{12C^P_t \gamma_t}$$

$$\le \frac{1}{12C^P_t \gamma_t} \left( \frac{\min\{\sqrt{V}, \log(Vt)\}}{1 - \sigma_2(P)} + 2 + D \right) \le \frac{1}{12\gamma_t}. \tag{25}$$

Where the last inequality comes from $D \ge 1$.

As mentioned before, for any fixed $k$ we have

$$\hat{L}_{u,t}(k) - \hat{L}^{obs}_{v,t}(k) \le V \sum_{\hat{t}=t-D}^{t-1} m_{\hat{t}}(k) + \|\hat{L}^{obs}_{v,t} - \bar{L}_t\|_\infty$$

$$\le V \sum_{\hat{t}=t-D}^{t-1} m_{\hat{t}}(k) + \frac{1}{12C^P_t \gamma_t} \left( \frac{\min\{\sqrt{V}, \log(Vt)\}}{1 - \sigma_2(P)} + 2 \right). \tag{26}$$

Using mathematical analysis, we assume that from $1$ to $t$ the lemma holds. Then using the same analytical method in Lemma 6, for any time step $t - D \le \hat{t} \le t - 1$, we can obtain:

$$\tilde{x}^k_{v,t} \le 2\tilde{x}^k_{u,\hat{t}} \le 4x^k_{u,\hat{t}}.$$

By Lemma 4 and mathematical induction, we have

$$\sum_{\hat{t}=t-D}^{t-1} \sum_{i=1}^{V} \sum_{k=1}^{K} \rho_t^k(x_{v,t}) m_{\hat{i}}(k) \leq \sum_{\hat{t}=t-D}^{t-1} \sum_{i=1}^{V} \sum_{k=1}^{K} \rho_t^k(x_{v,t}) \hat{\ell}_{i,\hat{t}}(k) \leq 8VD(K-1)^{\frac{1}{3}}. \quad (27)$$

According to our update rules for deviation records, no records generated in time period $[t-D+1, t]$ will be used, so we have

$$\hat{L}^{obs}_{v,t-D+1 \to t}(k) = V \sum_{\hat{t}=t-D}^{t-1} \sum_{u=1}^{V} P_{u,v}^{t-\hat{t}-1} \tilde{\ell}_{u,t}(k) + \tilde{\ell}_{v,t}(k) \leq \frac{D}{12C_t^P \gamma_t}. \quad (28)$$

Combine equation 22, equation 23 and equation 24, we can complete the right statement

$$\sum_{k=1}^{K} \rho_t^k(x_{v,t})(\hat{L}_{u,t}(k) - \hat{L}^{obs}_{v,t}(k))$$

$$\leq \frac{1}{12C_t^P \gamma_t} \left( \frac{\min\{\sqrt{V}, \log(Vt)\}}{1 - \sigma_2(P)} + 2 + D \right) + 8VD(K-1)^{\frac{1}{3}}$$

$$\leq \frac{1}{4\gamma_t}.$$

Where the last inequality uses the fact

$$\gamma_t^{-1} = 8V\sqrt{C_t^P t / \log K + 36D^2(K-1)^{2/3} + 4(C_t^P)^2} \geq 48Vd(K-1)^{\frac{1}{3}}.$$

By By Lemma 3, we can get

$$\tilde{x}_{u,t}^k \leq \frac{1}{1 - \frac{1}{12} - \frac{1}{4}} x_{v,t}^k \leq 2x_{v,t}^k.$$

$\square$

## 12.2 Adversarial Bounds

Then, the group regret can be bounded as follows:

$$\sum_{v=1}^{V} R_T(v) = \sum_{v=1}^{V} \mathbb{E} \left[ \sum_{t=1}^{T} \langle \bar{\ell}_t, x_{v,t} \rangle - \bar{\ell}_t(k^*) \right]$$

$$\overset{(a)}{=} \sum_{v=1}^{V} \mathbb{E} \left[ \sum_{t=1}^{T} (\langle \mathbb{E}[m_t], x_{v,t} \rangle - \mathbb{E}[m_t(k^*)]) \right]$$

$$\overset{(b)}{\leq} \mathbb{E} \left[ \sum_{v=1}^{V} \sum_{t=1}^{T} \langle m_t, x_{v,t} \rangle - \hat{L}_{1,T+1}(k^*) \right]$$

$$= \mathbb{E} \left[ \underbrace{\sum_{t=1}^{T} \sum_{v=1}^{V} \left( \bar{F}_t^*(-\hat{L}^{obs}_{v,t} - m_t) - \bar{F}_t^*(-\hat{L}^{obs}_{v,t}) + \langle x_{v,t}, m_t \rangle \right)}_{(A)}$$

$$+ \underbrace{\sum_{t=1}^{T} \sum_{v=1}^{V} \left( \bar{F}_t^*(-\hat{L}^{obs}_{v,t}) - \bar{F}_t^*(-\hat{L}^{obs}_{v,t} - m_t) - \bar{F}_t^*(-\hat{L}_{v,t}) + \bar{F}_t^*(-\hat{L}_{v+1,t}) \right)}_{(B)}$$

$$+ \underbrace{\left( \sum_{v=1}^{V} \sum_{t=1}^{T} \bar{F}_t^*(-\hat{L}_{v,t}) - \bar{F}_t^*(-\hat{L}_{v+1,t}) \right) - \hat{L}_{1,T+1}(k^*)}_{(C)} \right].$$

Where (a) holds because the following facts for all arms $k$:

$$\bar{\ell}_t(k) = \frac{1}{V}\sum_{v=1}^{V}\ell_{v,t}(k) = \frac{1}{V}\sum_{v=1}^{V}x_{v,t}(k)\frac{\ell_{v,t}(k)}{x_{v,t}(k)} = \mathbb{E}[m_t(k)],$$

(b) holds because the following definition:

$$\hat{L}_{1,T+1}(k^*) = \sum_{t=1}^{T}Vm_t(k^*) = \sum_{v=1}^{V}\sum_{t=1}^{T}m_t(k^*).$$

### 12.2.1 BOUNDING (A)

We introduce the count variable

$$Y_{k,t} \triangleq \sum_{v=1}^{V}\mathbb{I}\{k_{v,t} = k\},$$

which records how many agents select arm $k$ at round $t$. By this definition, for any arm $k$ we have

$$m_{k,t} \leq \frac{1}{V}\sum_{v:\,k_{v,t}=k}\frac{\ell_{v,t}(k)}{x_{v,t}(k)} \leq \frac{1}{V}\sum_{v:\,k_{v,t}=k}\frac{1}{x_{v,t}(k)} \leq \frac{3Y_{k,t}}{2V\,x_{v,t}(k)}, \tag{29}$$

where the first inequality uses $\ell_{v,t} \leq 1$, and the last inequality follows from Lemma 1.

$$\sum_{v=1}^{V}\bar{F}_t^*(-\hat{L}_{v,t}^{obs} - m_t) - \bar{F}_t^*(-\hat{L}_{v,t}^{obs}) + \langle x_{v,t}, m_t\rangle$$

$$\overset{(a)}{=} \sum_{v=1}^{V}\bar{F}_t^*(-\nabla\bar{F}_t(x_{v,t}) - m_t) - \bar{F}_t^*(-\nabla\bar{F}_t(x_{v,t})) + \langle x_{v,t}, m_t\rangle$$

$$\overset{(b)}{\leq} \sum_{v=1}^{V}F_t^*(-\nabla\bar{F}_t(x_{v,t}) - m_t) - F_t^*(-\nabla\bar{F}_t(x_{v,t})) + \langle x_{v,t}, m_t\rangle$$

$$= \sum_{v=1}^{V}\sum_{k=1}^{K}D_{f_t^*}\left(f'(x_{v,t}) - m_{k,t}, f'(x_{v,t})\right)$$

$$\overset{(c)}{=} \sum_{v=1}^{V}\frac{V}{Y_{k_{v,t},t}}D_{f_t^*}\left(f'(x_{v,t}(k_{v,t})) - m_{k_{v,t},t}, f'(x_{v,t}(k_{v,t}))\right)$$

$$\overset{(d)}{\leq} \sum_{v=1}^{V}\frac{V}{Y_{k_{v,t},t}}D_{f_t^*}\left(f'(x_{v,t}(k_{v,t})) - \frac{3Y_{k_{v,t},t}}{2V\,x_{v,t}(k_{v,t})}, f'(x_{v,t}(k_{v,t}))\right)$$

$$\overset{(e)}{\leq} \sum_{v=1}^{V}\frac{9Y_{k_{v,t},t}}{8V(x_{v,t}(k_{v,t}))^2 f_t''(x_{v,t}(k_{v,t}))}$$

$$\leq \frac{9}{8}\sum_{v=1}^{V}\frac{1}{(x_{v,t}(k_{v,t}))^2 f_t''(x_{v,t}(k_{v,t}))}$$

$$\leq \frac{9}{32}\sum_{v=1}^{V}\frac{x_{v,t}(k_{v,t})^{3/2}}{(x_{v,t}(k_{v,t}))^2\sqrt{Vt + 169V^2D}}$$

$$= \frac{9}{32}\sum_{v=1}^{V}\frac{1}{(x_{v,t}(k_{v,t}))^{\frac{1}{2}}\sqrt{Vt}}.$$

Where (a) applies Facts 2 and 3, the (b) follows from both parts of Fact 4, (d) holds because equation 29, (e) uses Fact 6, and (c) uses the following equality for any arm $k$:

$$\sum_{v=1}^{V}D_{f_t^*}\left(f'(x_{v,t}) - m_{k,t}, f'(x_{v,t})\right) = \frac{V}{Y_{k,t}}\sum_{k_{v,t}=k}D_{f_t^*}\left(f'(x_{v,t}) - m_{k,t}, f'(x_{v,t})\right).$$

Taking expectations and summing over $t \in [T]$ yields

$$\mathbb{E}\Big[ \sum_{t=1}^{T} \sum_{v=1}^{V} \big( \bar{F}_t^*(-L_{v,t}^{obs} - m_t) - \bar{F}_t^*(-L_{v,t}^{obs}) + \langle x_{v,t}, \tilde{\ell}_t \rangle \big) \Big]$$

$$\leq \frac{9}{32} \sum_{t=1}^{T} \sum_{v=1}^{V} \sum_{k=1}^{K} \frac{\sqrt{x_{v,t}(k)}}{\sqrt{Vt}} \leq \frac{9}{16} \sqrt{VKT}. \qquad (30)$$

where we use $\sum_{k=1}^{K} \sqrt{x_{v,t}(k)} \leq \sqrt{K}$ and $\sum_{t=1}^{T} t^{-\frac{1}{2}} \leq 2\sqrt{T}$.

### 12.2.2 BOUNDING (B)

We define $\hat{L}_{v,t}^{miss} = \hat{L}_{v,t} - \hat{L}_{v,t}^{obs}$. Then we have for any $v \in [V]$ and $t \in [T]$ :

$$-\bar{F}_t^*(-\hat{L}_{v,t}) + \bar{F}_t^*(-\hat{L}_{v+1,t}) = -\bar{F}_t^*(-\hat{L}_{v,t}) + \bar{F}_t^*(-\hat{L}_{v,t} - m_t)$$

$$= -\int_0^1 \langle m_t, \nabla \bar{F}_t^*(-\hat{L}_{v,t} - xm_t) \rangle dx$$

$$= -\int_0^1 \langle m_t, \nabla \bar{F}_t^*(-\hat{L}_{v,t}^{obs} - \hat{L}_{v,t}^{miss} - xm_t) \rangle dx.$$

Where the second equality holds by the fundamental theorem of calculus. Therefore, we have for any $v \in [V]$ and $t \in [T]$ :

$$\sum_{v=1}^{V} \bar{F}_t^*(-\hat{L}_{v,t}^{obs}) - \bar{F}_t^*(-\hat{L}_{v,t}^{obs} - m_t) - \bar{F}_t^*(-\hat{L}_{v,t}) + \bar{F}_t^*(-\hat{L}_{v+1,t})$$

$$\stackrel{(a)}{\leq} \sum_{v=1}^{V} \int_0^1 \langle m_t, \nabla \bar{F}_t^*(-\hat{L}_{v,t}^{obs} - xm_t) \rangle dx - \int_0^1 \langle m_t, \nabla \bar{F}_t^*(-\hat{L}_{v,t}^{obs} - \hat{L}_{v,t}^{miss} - xm_t \rangle dx$$

$$\stackrel{(b)}{=} \sum_{v=1}^{V} \sum_{k=1}^{K} \int_0^1 \langle m_{k,t}, \tilde{z}(x) - \nabla \bar{F}_t^*(\nabla F_t(\tilde{z}(x)) - \hat{L}_{v,t}^{miss}) \rangle dx$$

$$\stackrel{(c)}{\leq} \sum_{v=1}^{V} \sum_{k=1}^{K} \int_0^1 \langle m_{k,t}, \tilde{z}(x) - \nabla \bar{F}_t^*(\nabla F_t(\tilde{z}(x)) - \hat{L}_{v,t}^{miss}(k)) \rangle dx$$

$$\stackrel{(d)}{\leq} \sum_{v=1}^{V} \sum_{k=1}^{K} \int_0^1 \langle m_{k,t}, \tilde{z}(x) - \nabla F_t^*(\nabla F_t(\tilde{z}(x)) - \hat{L}_{v,t}^{miss}(k)) \rangle dx$$

$$= \sum_{v=1}^{V} \sum_{k=1}^{K} \int_0^1 m_{k,t}(\tilde{z}_k(x) - f_t^{*'}(f_t'(\tilde{z}_k(x)) - \hat{L}_{v,t}^{miss}(k)) dx$$

$$\stackrel{(e)}{\leq} \sum_{v=1}^{V} \sum_{k=1}^{K} \int_0^1 m_{k,t} f_t^{*''}(f_t'(\tilde{z}_k(x)) \hat{L}_{v,t}^{miss}(k) dx$$

$$= \sum_{v=1}^{V} \sum_{k=1}^{K} \int_0^1 m_{k,t} f_t''(\tilde{z}_k(x))^{-1} \hat{L}_{v,t}^{miss}(k) dx$$

$$\stackrel{(f)}{\leq} \sum_{v=1}^{V} \sum_{k=1}^{K} \int_0^1 m_{k,t} f_t'' \left( \frac{3}{2} x_{v,t}(k) \right)^{-1} \hat{L}_{v,t}^{miss}(k) dx$$

$$\leq \frac{3\gamma_t}{2} \sum_{v=1}^{V} \sum_{k=1}^{K} m_{k,t} x_{v,t}(k) \hat{L}_{v,t}^{miss}(k) dx$$

$$\stackrel{(g)}{\leq} \frac{9\gamma_t}{4V} \sum_{v=1}^{V} \sum_{u=1}^{V} \hat{L}_{v,t}^{miss}(k_{u,t}).$$

Where (a) uses the Fundamental theorem of calculus together with the inequality above, (b) substitutes $\tilde{z}(x) = \nabla \bar{F}_t(-\hat{L}_{v,t}^{obs} - xm_t)$ and applies Fact 3, (c) follows from the fact that $\nabla \bar{F}_t^*(-L)_k$ decreases if the loss in coordinates other than $k$ is reduced, (d) applies Fact 5, (e) $f_t^{*'}$ is convex, so $-f_t^{*'}(f_t'(\tilde{z}_k(x)) - \hat{L}_{v,t}^{miss}(k)) \le -\tilde{z}_k(x) + f_t^{*''}(f_t'(\tilde{z}_k(x))\hat{L}_{v,t}^{miss}(k)$, (f) follows because $\tilde{z}_k \le \frac{3}{2}x_{v,t}(k)$ and $F_t''(x)^{-1}$ is monotonically increasing, and (g) holds because the following inequality:

$$\sum_{v=1}^{V}\sum_{k=1}^{K} m_{k,t}x_{v,t}(k)\hat{L}_{v,t}^{miss}(k) = \frac{1}{V}\sum_{v=1}^{V}\sum_{k=1}^{K}\sum_{u=1}^{V} \mathbb{I}(k = k_{u,t})\hat{\ell}_{u,t}(k_{u,t})x_{v,t}(k)\hat{L}_{v,t}^{miss}(k)$$

$$= \frac{1}{V}\sum_{v=1}^{V}\sum_{u=1}^{V} \hat{\ell}_{u,t}(k_{u,t})x_{v,t}(k_{u,t})\hat{L}_{v,t}^{miss}(k_{u,t})$$

$$= \frac{1}{V}\sum_{v=1}^{V}\sum_{u=1}^{V} V\ell_{u,t}(k_{u,t})\hat{L}_{v,t}^{miss}(k_{u,t})\frac{x_{v,t}(k_{u,t})}{x_{u,t}(k_{u,t})}$$

$$\le \frac{3}{2V}\sum_{v=1}^{V}\sum_{u=1}^{V} \hat{L}_{v,t}^{miss}(k_{u,t}).$$

For any fixed $k, v, t$, we have

$$\mathbb{E}\left[\|\tilde{\ell}_{v,t}(k)\|_\infty\right] = \left\|\frac{\ell_{v,t}(k)x_{v,t}(k)}{\max\{x_{v,t}(k), 12C_t^P\gamma_t\}}\right\|_\infty \le 1,$$

and

$$\mathbb{E}\left[\|\hat{\ell}_{v,t}(k)\|_\infty\right] = \mathbb{E}\left[\mathbb{I}(k_{v,t} = k)\left\|\frac{\ell_{v,t}(k)}{x_{v,t}(k)}\right\|_\infty\right] = \left\|\frac{\ell_{v,t}(k)x_{v,t}(k)}{x_{v,t}(k)}\right\|_\infty \le 1,$$

Using the same analytical method in Lemma 6, we can obtain

$$\mathbb{E}\left[\hat{L}_{v,t}^{miss}(k_{u,t})\right] \le \mathbb{E}\left[\left\|\hat{L}_{v,1\to t-D} - \hat{L}_{v,1\to t-D}^{obs}\right\|_\infty + \left\|\hat{L}_{v,t-D+1\to t} - \hat{L}_{v,t-D+1\to t}^{obs}\right\|_\infty\right]$$

$$\le V\left(\frac{\min\{\sqrt{V}, \log(Vt)\}}{1 - \sigma_2(P)} + 2\right) + VD = VC_t^P.$$

Finally, we have in expectation

$$\mathbb{E}\left[\sum_{t=1}^{T}\sum_{v=1}^{V} \left(\bar{F}_t^*(-\hat{L}_{v,t}^{obs}) - \bar{F}_t^*(-\hat{L}_{v,t}^{obs} - m_t) - \bar{F}_t^*(-\hat{L}_{v,t}) + \bar{F}_t^*(-\hat{L}_{v+1,t})\right)\right]$$

$$\le \sum_{t=1}^{T}\frac{9\gamma_t}{4V}\mathbb{E}\left[\sum_{v=1}^{V}\sum_{u=1}^{V}\hat{L}_{v,t}^{miss}(k_{u,t})\right]$$

$$\le \sum_{t=1}^{T}\frac{9\gamma_t}{4V}\cdot V^3 C_t^P$$

$$\le \frac{9}{16}V\sqrt{C_T^P T \log K}. \tag{31}$$

### 12.2.3 BOUNDING (C)

Let

$$\tilde{x}_{1,t} \in \arg\max_{x\in\Delta^{K-1}} \langle x, -\hat{L}_{1,t}\rangle - F_t(x).$$

By the definition of the convex conjugate, we have

$$\bar{F}_t^*(-\hat{L}_{1,t}) = \langle \tilde{x}_{1,t}, -\hat{L}_{1,t}\rangle - F_t(\tilde{x}_{1,t}).$$

Moreover, since $\bar{F}_{t-1}^*(-\hat{L}_{1,t}) = \max_{x \in \Delta^{K-1}} \langle x, -\hat{L}_{1,t} \rangle - F_{t-1}(x)$, evaluating the maximization at $\tilde{x}_{1,t}$ yields

$$-\bar{F}_{t-1}^*(-\hat{L}_{1,t}) \ \leq \ -\langle \tilde{x}_{1,t}, -\hat{L}_{1,t} \rangle + F_{t-1}(\tilde{x}_{1,t}). \tag{32}$$

Similarly, we have

$$-\bar{F}_T^*(-\hat{L}_{1,T+1}) \ \leq \ -\langle e_{k^*}, -\hat{L}_{1,T+1} \rangle + F_T(e_{k^*}) \ \leq \ \hat{L}_{1,T+1}(k^*), \tag{33}$$

Plugging equation 32–equation 33 into the left-hand side, and using the telescoping identity

$$\sum_{t=1}^{T} \sum_{v=1}^{V} \left( \bar{F}_t^*(-\hat{L}_{v,t}) - \bar{F}_t^*(-\hat{L}_{v+1,t}) \right) = \sum_{t=1}^{T} \left( \bar{F}_t^*(-\hat{L}_{1,t}) - \bar{F}_t^*(-\hat{L}_{1,t+1}) \right),$$

we obtain

$$\sum_{t=1}^{T} \left( \sum_{v=1}^{V} \left( \bar{F}_t^*(-\hat{L}_{v,t}) - \bar{F}_t^*(-\hat{L}_{v+1,t}) \right) \right) - \hat{L}_{1,T+1}(k^*)$$

$$= \sum_{t=1}^{T} \left( \bar{F}_t^*(-\hat{L}_{1,t}) - \bar{F}_t^*(-\hat{L}_{1,t+1}) \right) - \hat{L}_{1,T+1}(k^*)$$

$$\leq -F_1(\tilde{x}_{1,1}) + \sum_{t=2}^{T} \left( F_{t-1}(\tilde{x}_{1,t}) - F_t(\tilde{x}_{1,t}) \right)$$

$$\leq \max_{x \in \Delta^{K-1}} \left( -F_1(x) \right) + \sum_{t=2}^{T} \max_{x \in \Delta^{K-1}} \left( F_{t-1}(x) - F_t(x) \right)$$

$$= -F_T(\mathbf{1}_K/K)$$

$$= 8\sqrt{VKT + 169V^2 D} + 8V \log K \sqrt{C_T^P T / \log K + 36D^2(K-1)^{\frac{2}{3}} + 4(C_t^P)^2}$$

$$\leq 8\sqrt{VKT} + 8V\sqrt{C_T^P T \log K} + 104V\sqrt{D} + 48VD(K-1)^{\frac{1}{3}} \log K + 16VC_T^P \log K. \tag{34}$$

Combine equation 30, equation 31, and equation 34, we can get

$$\sum_{v=1}^{V} R_T(v) \leq$$
$$\frac{137}{16} \sqrt{VKT} + \frac{137}{16} V \sqrt{C_T^P T \log K} + 104V\sqrt{D} + 48VD(K-1)^{\frac{1}{3}} \log(K + 16VC_T^P \log K.$$

Moreover, Lemma 1 implies that the individual regret is controlled by the group regret:

$$R_T(v) \leq \frac{3}{2V} \sum_{v=1}^{V} R_T(v)$$

$$< 13\sqrt{KT/V} + 13\sqrt{C_T^P T \log K} + 156\sqrt{D} + 72D(K-1)^{\frac{1}{3}} \log K + 24C_T^P \log K.$$

## 12.3 STOCHASTIC BOUNDS

Inspired the analysis of stochastic bound for bandit with delay feedback in Masoudian et al. (2022), let $\tilde{x}_{v,t} = \nabla \bar{F}_t^*(-\hat{L}_{v,t})$, then we define the drifted pseudo-regret as

$$R_T^{drift}(v) = \mathbb{E}\left[ \sum_{t=1}^{T} \left( \langle \tilde{x}_{v,t}, \bar{\ell}_t \rangle - \bar{\ell}_t(k^*) \right) \right].$$

We rewrite the drifted regret as

$$R_T^{drift}(v) = \mathbb{E}\left[ \sum_{t=1}^{T} \left( \langle \tilde{x}_{v,t}, \bar{\ell}_t \rangle - \bar{\ell}_t(k^*) \right) \right] = \sum_{t=1}^{T} \sum_{k=1}^{K} \mathbb{E}\left[ \left( \tilde{x}_{v,t}^k, \bar{\ell}_{k,t} - \bar{\ell}_t(k^*) \right) \right] = \sum_{t=1}^{T} \sum_{k=1}^{K} \mathbb{E}[\tilde{x}_{v,t}^k] \Delta_k.$$

Using the Lemma 7, for any agent $v$ we have

$$\frac{5}{3}R_T^{drift}(v) = \frac{5}{3}\sum_{t=1}^{T}\sum_{k=1}^{K}\mathbb{E}[\tilde{x}_{v,t}^k]\Delta_k \geq \sum_{t=1}^{T-D}\sum_{k=1}^{K}\mathbb{E}[x_{v,t+D}^k]\Delta_k$$

$$= \sum_{t=D+1}^{T}\sum_{k=1}^{K}\mathbb{E}[x_{v,t}^k]\Delta_k$$

$$\geq \sum_{t=1}^{T}\sum_{k=1}^{K}\mathbb{E}[x_{v,t}^k]\Delta_k - D$$

$$= R_T(v) - D.$$

Where the second inequality uses $\sum_{t=1}^{D}\sum_{k=1}^{K}\mathbb{E}[x_{v,t}^k]\Delta_k \leq D$. As a result, we have $R_T(v) \leq \frac{5}{3}R_T^{drift}(v) + D$ and it suffices to upper bound $R_T^{drift}(v)$. As a consequence, the drifted pseudo-regret bound as follows:

$$\sum_{v=1}^{V}R_T^{drift}(v) = \sum_{v=1}^{V}\mathbb{E}\left[\sum_{t=1}^{T}\langle\bar{\ell}_t, \tilde{x}_{v,t}\rangle - \bar{\ell}_t(k^*)\right]$$

$$= \mathbb{E}\left[\sum_{v=1}^{V}\sum_{t=1}^{T}\langle\bar{\ell}_t, \tilde{x}_{v,t}\rangle - \bar{\ell}_t(k^*)\right]$$

$$\overset{(a)}{=} \mathbb{E}\left[\sum_{v=1}^{V}\sum_{t=1}^{T}\left(\langle\mathbb{E}[m_t], \tilde{x}_{v,t}\rangle - \mathbb{E}[m_t(k^*)]\right)\right]$$

$$\overset{(b)}{\leq} \mathbb{E}\left[\sum_{v=1}^{V}\sum_{t=1}^{T}\langle m_t, \tilde{x}_{v,t}\rangle - \hat{L}_{1,T+1}(k^*)\right]$$

$$= \mathbb{E}\left[\underbrace{\sum_{t=1}^{T}\sum_{v=1}^{V}\left(\bar{F}_t^*(-\hat{L}_{v+1,t}) - \bar{F}_t^*(-\hat{L}_{v,t}) + \langle\tilde{x}_{v,t}, m_t\rangle\right)}_{(A)}\right.$$

$$\left. + \underbrace{\left(\sum_{t=1}^{T}\sum_{v=1}^{V}\bar{F}_t^*(-\hat{L}_{v,t}) - \bar{F}_t^*(-\hat{L}_{v+1,t})\right) - \hat{L}_{1,T+1}(k^*)}_{(B)}\right].$$

Where (a) holds because the following facts for all arms $k$:

$$\bar{\ell}_t(k) = \frac{1}{V}\sum_{v=1}^{V}\ell_{v,t}(k) = \frac{1}{V}\sum_{v=1}^{V}x_{v,t}(k)\frac{\ell_{v,t}(k)}{x_{v,t}(k)} = \mathbb{E}[m_{v,t}(k)],$$

(b) holds because the following definition:

$$\hat{L}_{1,T+1}(k^*) = \sum_{t=1}^{T}Vm_t(k^*) = \sum_{v=1}^{V}\sum_{t=1}^{T}m_t(k^*).$$

### 12.3.1 BOUNDING (A)

$$\sum_{v=1}^{V}\bar{F}_t^*(-\hat{L}_{v+1,t}) - \bar{F}_t^*(-\hat{L}_{v,t}) + \langle\tilde{x}_{v,t}, m_t\rangle$$

$$= \sum_{v=1}^{V}\bar{F}_t^*(-\hat{L}_{v,t} - m_t) - \bar{F}_t^*(-\hat{L}_{v,t}) + \langle\tilde{x}_{v,t}, m_t\rangle$$

$$= \sum_{v=1}^{V} \bar{F}_t^*(-\hat{L}_{v,t} - (m_t - \tilde{x}_{v,t} \odot m_t)) - \bar{F}_t^*(-\hat{L}_{v,t}) + \langle \tilde{x}_{v,t}, m_t - \tilde{x}_{v,t} \odot m_t \rangle$$

$$\overset{(a)}{=} \sum_{v=1}^{V} \bar{F}_t^*(-\nabla \bar{F}_t(\tilde{x}_{v,t}) - (m_t - \tilde{x}_{v,t} \odot m_t)) - \bar{F}_t^*(-\nabla \bar{F}_t(x_{v,t})) + \langle \tilde{x}_{v,t}, m_t - \tilde{x}_{v,t} \odot m_t \rangle$$

$$\overset{(b)}{\le} \sum_{v=1}^{V} F_t^*(-\nabla \bar{F}_t(\tilde{x}_{v,t}) - (m_t - \tilde{x}_{v,t} \odot m_t)) - F_t^*(-\nabla \bar{F}_t(\tilde{x}_{v,t})) + \langle \tilde{x}_{v,t}, m_t - \tilde{x}_{v,t} \odot m_t \rangle$$

$$= \sum_{v=1}^{V} \sum_{k=1}^{K} D_{f_t^*}\left( f'(\tilde{x}_{v,t}) - (m_t(k) - \tilde{x}_{v,t}^k m_t(k)), f'(\tilde{x}_{v,t}) \right)$$

$$= \sum_{v=1}^{V} \sum_{k=1}^{K} D_{f_t^*}\left( f'(\tilde{x}_{v,t}) - \frac{1}{V} \sum_{u=1}^{V} \frac{\ell_{u,t}(1 - \tilde{x}_{v,t}^k)}{x_{u,t}^k}, f'(\tilde{x}_{v,t}) \right)$$

$$\le \sum_{v=1}^{V} \sum_{k=1}^{K} D_{f_t^*}\left( f'(\tilde{x}_{v,t}) - \frac{1}{V} \sum_{u=1}^{V} \frac{1 - \tilde{x}_{v,t}^k}{x_{u,t}^k}, f'(\tilde{x}_{v,t}) \right)$$

$$\overset{(c)}{\le} \sum_{v=1}^{V} \sum_{k=1}^{K} \frac{f_t''(\tilde{x}_{v,t}(k))^{-1}}{2V^2} \left( \sum_{u=1}^{V} \frac{1 - \tilde{x}_{v,t}^k}{x_{u,t}^k} \right)^2$$

$$\overset{(d)}{\le} \sum_{v=1}^{V} \sum_{k=1}^{K} \frac{\tilde{x}_{v,t}(k)^{\frac{3}{2}}}{8V^2\sqrt{Vt}} \left( 2V \frac{1 - \tilde{x}_{v,t}^k}{\tilde{x}_{v,t}^k} \right)^2$$

$$= \sum_{v=1}^{V} \sum_{k=1}^{K} \frac{(1 - \tilde{x}_{v,t}^k)^2}{2(\tilde{x}_{v,t}^k)^{\frac{1}{2}}\sqrt{Vt}}.$$

Where (a) applies Facts 2 and 3, the (b) follows from both parts of Fact 4, (c) uses Fact 6, and (d) uses the Lemma 8. In expectation we get

$$\mathbb{E}\left[ \sum_{v=1}^{V} \left( \bar{F}_t^*(-\hat{L}_{v+1,t}) - \bar{F}_t^*(-\hat{L}_{v,t}) + \langle \tilde{x}_{v,t}, m_t \rangle \right) \right]$$

$$\le \sum_{v=1}^{V} \sum_{k=1}^{K} \frac{(1 - \tilde{x}_{v,t}^k)^2 (\tilde{x}_{v,t}^k)^{\frac{1}{2}}}{2\sqrt{Vt}}.$$

$$\le \sum_{v=1}^{V} \sum_{k \neq k^*} \frac{(\tilde{x}_{v,t}^k)^{\frac{1}{2}}}{\sqrt{Vt}} + \sum_{v=1}^{V} \frac{(1 - \tilde{x}_{v,t}(k^*))^2 (\tilde{x}_{v,t}(k^*))^{\frac{1}{2}}}{2\sqrt{Vt}}$$

$$\le \sum_{v=1}^{V} \sum_{k \neq k^*} \frac{(\tilde{x}_{v,t}^k)^{\frac{1}{2}}}{\sqrt{Vt}} + \sum_{v=1}^{V} \frac{(1 - \tilde{x}_{v,t}(k^*))}{2\sqrt{Vt}}$$

$$= \sum_{v=1}^{V} \sum_{k \neq k^*} \frac{(\tilde{x}_{v,t}^k)^{\frac{1}{2}}}{2\sqrt{Vt}} + \sum_{v=1}^{V} \sum_{k \neq k^*} \frac{\tilde{x}_{v,t}^k}{2\sqrt{Vt}}$$

$$\le \sum_{v=1}^{V} \sum_{k \neq k^*} \frac{(\tilde{x}_{v,t}^k)^{\frac{1}{2}}}{\sqrt{Vt}}. \tag{35}$$

### 12.3.2 BOUNDING (B)

For any $u \in [V]$ we have

$$\left( \sum_{t=1}^{T} \sum_{v=1}^{V} \bar{F}_t^*(-\hat{L}_{v,t}) - \bar{F}_t^*(-\hat{L}_{v+1,t}) \right) = \left( \sum_{t=1}^{T} \bar{F}_t^*(-\hat{L}_{v,t}) - \bar{F}_{t+1}^*(-\hat{L}_{v+1,t}) \right) - F_1(-\hat{L}_{v,1})$$

We have the following bound by Abernethy et al. (2015)

$$\big(\sum_{t=1}^{T}\sum_{v=1}^{V}\bar{F}_t^*(-\hat{L}_{v,t}) - \bar{F}_t^*(-\hat{L}_{v+1,t})\big) - \hat{L}_{1,T+1}(k^*)$$

$$= \frac{1}{V}\big(\sum_{v=1}^{V}\sum_{t=1}^{T}\bar{F}_t^*(-\hat{L}_{v,t}) - \bar{F}_t^*(-\hat{L}_{v,t+1})\big) - \hat{L}_{1,T+1}(k^*)$$

$$\le \frac{1}{V}\sum_{v=1}^{V}\sum_{t=2}^{T}\big((F_{t-1}(\tilde{x}_{v,t})) - (F_t(\tilde{x}_{v,t}))\big) + F_T(x^*) - \frac{1}{V}\sum_{v=1}^{V}F_1(\tilde{x}_{v,1}).$$

which gives us

$$\big(\sum_{t=1}^{T}\sum_{v=1}^{V}\bar{F}_t^*(-\hat{L}_{v,t}) - \bar{F}_t^*(-\hat{L}_{v+1,t})\big) - \hat{L}_{1,T+1}(k^*)$$

$$\le \frac{1}{V}\sum_{v=1}^{V}\sum_{t=2}^{T}\Big(2\sum_{k\neq k^*}(\tilde{x}_{v,t}^k)^{\frac{1}{2}}\big(\eta_t^{-1} - \eta_{t-1}^{-1}\big) - \sum_{k=1}^{K}\tilde{x}_{v,t}^k\log(\tilde{x}_{v,t}^k)\big(\gamma_t^{-1} - \gamma_{t-1}^{-1}\big)\Big)$$

$$+ 2\sqrt{\eta_0(K-1)} + \sqrt{\gamma_0\log K}$$

$$\le \sum_{v=1}^{V}\sum_{t=2}^{T}\Big(2\sum_{k\neq k^*}\eta_t(\tilde{x}_{v,t}^k)^{\frac{1}{2}} - \sum_{k=1}^{K}\frac{C_t^P\gamma_t\tilde{x}_{v,t}^k\log(\tilde{x}_{v,t}^k)}{\sqrt{\log K}}\Big) + 2\sqrt{\eta_0(K-1)} + \sqrt{\gamma_0\log K}. \quad (36)$$

where the first inequality holds because $x_{v,t}^{\frac{1}{2}}(k^*) \le 1$ and the second inequality follows by

$$\eta_t^{-1} - \eta_{t-1}^{-1} \le V\eta_t \quad and \quad \gamma_t^{-1} - \gamma_{t-1}^{-1} = \frac{\gamma_t^{-2} - \gamma_{t-1}^{-2}}{\gamma_t^{-1} - \gamma_{t-1}^{-1}} \le \frac{VC_t^P\gamma_t}{\sqrt{\log K}}$$

Combine equation 35 and equation 36, we can get

$$\sum_{v=1}^{V}R_T^{drift}(v) \le \sum_{t=2}^{T}\sum_{v=1}^{V}\sum_{k\neq k^*}\frac{3(\tilde{x}_{v,t}^k)^{\frac{1}{2}}}{2\sqrt{Vt}} + \sum_{t=2}^{T}\sum_{v=1}^{V}\sum_{k=1}^{K}\frac{C_t^P\gamma_{t-1}\tilde{x}_{v,t}^k\log(1/\tilde{x}_{v,t}^k)}{\log K}$$

$$+ 22VD\sqrt{(K-1)\log K} + 4VC_T^P\log K. \quad (37)$$

### 12.3.3 SELF BOUNDING ANALYSIS

We use the self-bounding technique to write $\sum_{v=1}^{V}R_T^{drift}(v) = 3\sum_{v=1}^{V}R_T^{drift}(v) - 2\sum_{v=1}^{V}R_T^{drift}(v)$, and then based on equation 37 we have

$$\sum_{v=1}^{V}R_T^{drift}(v) \le \sum_{v=1}^{V}\sum_{k\neq k^*}\frac{9(\tilde{x}_{v,t}^k)^{\frac{1}{2}}}{2\sqrt{Vt}} - \sum_{v=1}^{V}R_T^{drift}(v)$$

$$+ \sum_{t=2}^{T}\sum_{v=1}^{V}\sum_{k=1}^{K}\frac{3C_t^P\gamma_{t-1}\tilde{x}_{v,t}^k\log(1/\tilde{x}_{v,t}^k)}{\log K} - \sum_{v=1}^{V}R_T^{drift}(v)$$

$$+ 22VD\sqrt{(K-1)\log K} + 4VC_T^P\log K.$$

Here we give bound for the first term:

$$\sum_{v=1}^{V}\sum_{k\neq k^*}\frac{9(\tilde{x}_{v,t}^k)^{\frac{1}{2}}}{2\sqrt{Vt}} - \sum_{v=1}^{V}R_T^{drift}(v) = \sum_{t=1}^{T}\sum_{v=1}^{V}\sum_{k\neq k^*}\Big(\frac{9(\tilde{x}_{v,t}^k)^{4\frac{1}{2}}}{2\sqrt{Vt}} - \tilde{x}_{v,t}^k\Delta_k\Big)$$

$$\le \sum_{t=1}^{T}\sum_{v=1}^{V}\sum_{k\neq k^*}\frac{81}{4Vt\Delta_k} \le \sum_{k\neq k^*}\frac{81\log T}{4\Delta_k}.$$

where the first inequality uses $\forall x, y \geq 0 : x + y \geq 2\sqrt{xy} \Rightarrow 2\sqrt{xy} - y \leq x$ so called AM-GM.

According the proof of Lemma 8 in Masoudian et al. (2022), we can get bound for the second term:

$$\sum_{t=2}^{T}\sum_{v=1}^{V}\sum_{k=1}^{K}\frac{3C_t^P \gamma_{t-1}\tilde{x}_{v,t}^k \log(1/\tilde{x}_{v,t}^k)}{\log K} - \sum_{v=1}^{V} R_T^{drift}(v) \leq \sum_{k \neq k^*}\frac{36VC_T^P}{\Delta_k \log K}.$$

In summary, we have

$$\sum_{v=1}^{V} R_T(v) \leq \frac{5}{3}\sum_{v=1}^{V} R_T^{drift}(v) + VD$$

$$\leq \sum_{k \neq k^*}\frac{34\log T}{\Delta_k} + \sum_{k \neq k^*}\frac{60VC_T^P}{\Delta_k \log K} + 37VD\sqrt{(K-1)\log K} + 7VC_T^P \log K + VD.$$

For any $k, v, t$, by Lemma 1, we can get the individual regret for each agent $v$:

$$R_T(v) \leq \frac{3}{2V}\sum_{v=1}^{V} R_T(v)$$

$$\leq \sum_{k \neq k^*}\frac{51\log T}{V\Delta_k} + \sum_{k \neq k^*}\frac{90C_T^P}{\Delta_k \log K} + 56D\sqrt{(K-1)\log K} + 11C_T^P \log K + 2D.$$

### 12.4 PROOF FOR COMMUNICATION COST

At round $t$, the message sent by agent $v$ consists of two parts: (i) $\hat{L}_{v,t}^{obs}$ and (ii) the deviation-record set $A_v$. The size of $\hat{L}_{v,t}^{obs}$ is $O(K)$. Moreover, each deviation record remains in the system for at most $D$ rounds, so $A_v$ contains at most the records generated by all agents over the window $t - D < s \leq t$. Thus, we obtain

$$\mathbb{E}[Comm\_size(v,t)] = O(K) + O\left(\sum_{s=t-D+1}^{t}\sum_{i=1}^{V} i(12VC_t^P \gamma_t)^i\right)$$

$$= O(K) + O\left(\sum_{i=1}^{VD} i(12VC_t^P \gamma_t)^i\right)$$

$$= O(K) + O\left(\sum_{i=1}^{\infty} i(12VC_t^P \gamma_t)^i\right)$$

$$= O(K + \frac{12VC_t^P \gamma_t}{(1 - 12VC_t^P \gamma_t)^2}).$$

Where the last equality comes form the inequality:

$$\sum_{i=1}^{\infty} ia^i = \frac{a}{(1-a)^2}.$$

Finally, using $12VC_t^P \gamma_t \leq 0.75$, we conclude that $\mathbb{E}[comm\_cost(v,t)] = O(K)$.

