# OpenReview forum: "A Near-Optimal Best-of-Both-Worlds Algorithm for Federated Bandits"
_ICLR.cc/2026/Conference — ICLR 2026 Poster_

### Official Review · Reviewer_ouUu · 2025-10-30

**Soundness:** 3
**Presentation:** 3
**Contribution:** 2
**Rating:** 2
**Confidence:** 5

**Summary:**

This paper studies distributed multi-armed bandits with heterogeneous losses under both stochastic and adversarial regimes. It introduces FEDFTRL, a novel algorithm that is the first to achieve near-optimal regret in both settings. Comprehensive experiments are provided to validate the theoretical guarantees.

**Strengths:**

### Contributions
1. First establish an $O(\sqrt{T})$ regret bound in the adversarial regime.
2. Prove a best-of-both-worlds guarantee
3. Paper is well written.

**Weaknesses:**

1. Requires $O(K + VD)$ bits of communication per round; the protocol differs substantially from prior work (previous only required $O(K)$ ), so direct comparisons are not straightforward.
2. With an additional $O(VD)$ budget per round, the problem essentially reduces to a standard multi-armed bandit with delayed feedback.

**Questions:**

Please explain weaknesses, and I may significantly revise my review comments accordingly.

---

> ### Author Response · Authors · 2025-11-26
>
> Thank you for the insightful review. Here are our response to your comments.
>
> > Requires $O(K+VD)$ bits of communication per round
>
> In the revision, we reduced the expected communication cost from $O(K+VD)$ to $O(K)$ by slightly increasing the learning rate $\gamma_t$ from $\gamma_t^{-1} = 8V\sqrt{(C_t^P t)/\log K + 36D^2 (K-1)^{\frac{2}{3}}}$ to $\gamma_t^{-1} = 8V\sqrt{(C_t^P t)/\log K + 36D^2 (K-1)^{\frac{2}{3}} + 4(C_t^P)^2}$. This change only introduces an additional additive term $O(C_T^P \log K)$ in the regret bound and does not affect its asymptotic order. The numerical results of the new learning rate is almost the same as that of old one. We have updated our Theorem 1 and its proof in the appendix. The proof for the communication-cost is provided in section 11.4 in appendix.
>
> > With an additional $O(VD)$ budget per round, the problem essentially reduces to a standard multi-armed bandit with delayed feedback.
>
> Even with an additional $O(VD)$ budget, the studied problem does not reduces to a standard MAB with delayed feedback, as the gossip-based communication always introduces a consensus error. Consequently, each agent cannot obtain the exact global feedback required in standard MAB settings.

---

### Official Review · Reviewer_yECc · 2025-10-31

**Soundness:** 3
**Presentation:** 2
**Contribution:** 2
**Rating:** 4
**Confidence:** 3

**Summary:**

This work addresses the federated multi armed bandit problem and focuses on the heterogenous setting where agents cannot determine the globally optimal arm using their local biased observations. The major contribution is the proposed Best-of-Both-Worlds algorithm that performs robustly in both stochastic and adversarial environments. The proposed algorithm adapts the Follow-the-Regularized-Leader framework and incorporate a hybrid regularizer typically used for bandits with delayed feedback. The authors view this as analogous to the latency caused by decentralized communication. Novelty includes the use of a communication scheme that tracks ddeviation records and a truncated loss estiator to keep agent action probabilities nearly aligned despite the heterogeneous feedback. They demonstrate that the FEDFTRL algorithm achieves near optimal regret bounds.

**Strengths:**

1. The core contribution of the paper is the first Best-of-Both-Worlds regret guarantee for the heterogeneou federated bandit setting.

2. Achieving $O(\sqrt{T})$ individual regret in adversarial setting is a clear improvement over previous results.

3. The theoretical analysis seems deep and the derived regret bounds match the known lower bounds.

4. Experiments are comprehensive.

**Weaknesses:**

1. The major limitation is the communication complexity which is explicitly mentioned by the authors as well. Each agent requires communicating $O(K+VD)$ bits of information every round. This is a potential practical bottleneck for large scale federated systems with many agents or high network diameter.

2. The algorithm relies on special hybrid regularizers and fine tuned time varying learning rates that are defined based on network characteristics. Even though they are necessary for theoretical guarantees, this complexity may hinder deployment and tuning in practice.

3. While the adaptation is novel for federated setting, the theoretical framework relies heavily on importing and combining existing BOBW literature.

**Questions:**

1. Given that the communication cost is $O(K + V D)$ bits per round for each agent, can youprovide a more detailed discussion on the practical implications of this dependency on the number of agents $V$ and the network diameter $D$ for typical federated systems? Quantifying how $V$ and $D$ affect runtime in the experiments would be beneficial.

2. The parameter $C_t^P$ in Eq. (2) quantifies the delay caused by decentralized communication and $C_T^P$ captures the dependence on network topology. Can the authors further clarify the intuitive meaning of how these complexity measures dictate the regret?

3. The truncated loss estimator $\tilde{\ell}_{v,t}$ is crucial for stabilizing action probabilities. Can the authors comment on the practical robustness of the truncation threshold $(12V C_t^P \gamma_t)$ of the denominator. Especially, regarding its potential sensitivity to misspecified initial parameters or dynamic changes in the network topology captured by $C_t^P$?

---

> ### Author Response · Authors · 2025-11-26
>
> Thank you for the insightful review. Here are our response to your comments.
>
> > The major limitation is the communication complexity which is explicitly mentioned by the authors as well. Each agent requires communicating $O(K+VD)$ bits of information every round.
>
> In the revision, we reduced the expected communication cost from $O(K+VD)$ to $O(K)$ by slightly increasing the learning rate $\gamma_t$ from $\gamma_t^{-1} = 8V\sqrt{(C_t^P t)/\log K + 36D^2 (K-1)^{\frac{2}{3}}}$ to $\gamma_t^{-1} = 8V\sqrt{(C_t^P t)/\log K + 36D^2 (K-1)^{\frac{2}{3}} + 4(C_t^P)^2}$. This change only introduces an additional additive term $O(C_T^P \log K)$ in the regret bound and does not affect its asymptotic order. The numerical results of the new learning rate is almost the same as that of old one. We have updated our Theorem 1 and its proof in the appendix. The proof for the communication-cost is provided in section 11.4 in appendix.
>
> > The algorithm relies on special hybrid regularizers and fine tuned time varying learning rates that are defined based on network characteristics. Even though they are necessary for theoretical guarantees, this complexity may hinder deployment and tuning in practice.
>
> The hybrid regularizers are widely used in many previous works on  best-of-both-worlds bandit algorithms (Zimmert & Seldin, 2020; Masoudian et al., 2022; Ito, 2021a; Tsuchiya et al., 2023b). We believe that identifying a simple regularizer for federated bandits is an interesting problem and we will explore this in future work.
>
> > While the adaptation is novel for federated setting, the theoretical framework relies heavily on importing and combining existing BOBW literature.
>
> Though our work borrow some ideas from existing BOBW literature, our technical contributions are still significant:
>
> * Note that decentralized communication may introduce latency, as agents can only exchange information with their neighbors. We treat this latency as a form of feedback delay and introduce a novel time-varying parameter $C_T^P$ to quantify the delay induced by decentralized communication.
>
> * To address the issue of locally biased loss feedback, we employ a novel truncated loss estimator that stabilizes per-round updates and keeps the agents’ action distributions nearly aligned, while ensuring that the aggregated loss estimate at each round remains close to the average loss. In addition, we introduce a deviation-record correction scheme that compensates for the bias introduced by truncation, thereby further reducing regret.
>
> * Instead of directly upper-bounding individual regret as in Yi & Vojnović (2023), we first establish an upper bound on the group regret with respect to the global optimum. Leveraging the property that the agents’ action probabilities are nearly aligned, we then approximately divide this group regret by the number of agents to obtain the upper bound of  individual regret. This yields a more fine-grained and tight result.
>
> > The parameter $C_T^P$ in Eq. (2) quantifies the delay caused by decentralized communication and $C_T^P$ captures the dependence on network topology. Can the authors further clarify the intuitive meaning of how these complexity measures dictate the regret?
>
> Intuitively, a dense communication graph typically leads to a small $C_t^P$, which means that agents can reach consensus quickly. In this case, the regret of federated bandits approaches that of the centralized case. Conversely, if the communication graph is sparse, the parameter $C_t^P$ can become very large, and the additional term $\sqrt{C_T^P T \log(K)}$ in the adversarial regret may grow significantly.
>
> > The truncated loss estimator $\tilde \ell_{v,t}$ is crucial for stabilizing action probabilities. Can the authors comment on the practical robustness of the truncation threshold $12VC_t^P\gamma_t$ of the denominator. Especially, regarding its potential sensitivity to misspecified initial parameters or dynamic changes in the network topology captured by $C_t^P$ ?
>
> We have added robustness experiments for $C_t^P$, presented in Section 12 in appendix. The results show that our FedFTRL algorithm is robust to the choice of the topology parameter $C_t^P$.

---

### Official Review · Reviewer_cGPZ · 2025-11-01

**Soundness:** 3
**Presentation:** 3
**Contribution:** 3
**Rating:** 6
**Confidence:** 3

**Summary:**

The paper study the Best-of-Both-Worlds (BoBW) problem in decentralized federated bandit setting, where it follows the FTRL algorithm’s idea but handle delay and client heterogeneity in federated settings via a modified learning-rate schedule and truncated loss estimators. Theoretical results are provided for the proposed algorithm and the regret almost matches the lower bound in this setting. Numerical experiments validate the effectiveness of the proposed method.

**Strengths:**

Theoretical guarantee for the proposed method is strong which almost matches the lower bound in this setting.

**Weaknesses:**

We need to know the topology and $D$ beforehand.

**Questions:**

**Learning rates:** How are learning rates in (201) compared with that in the delayed feedback (Masoudian et al., 2022) ?

**Communication cost:** How is $x_{v,t}(k)$ compared to  $12VC_t^{P}\gamma_t$ in (4)? In practice, how many rounds do we need to truncate the loss and broadcast?

**equation (5):** It seems V multiplies to the loss after communication as well? In this case given $P$ is doubly stochastic, (when the loss estimates of the neighbors are closed), loss estimator seems to have exponential growth as time $t$ increases. Could the author elaborate this more?

**Clarification on matching bounds:** If the graph $G$ is known, could we always construct a doubly stochastic matrix $P$ as in Remark 1 to achieve the nearly matching bounds?

Typos:
line 155: (u,v)\not\in E

---

> ### Author Response · Authors · 2025-11-26
>
> Thank you for the insightful review. Here are our response to your comments.
>
> > We need to know the topology and $D$ beforehand.
>
> Most prior work on federated bandits (Zhu et al., 2021; Yi & Vojnović, 2023) and decentralized optimization (Yang et al., 2022; Koloskova et al., 2021) requires knowledge of the communication graph topology, as it is crucial for the gossip-based communication protocol.
>
> > Learning rates: How are learning rates in (line 201) compared with that in the delayed feedback (Masoudian et al., 2022) ?
>
> The learning rates in the delayed feedback are $\eta_t^{-1} = \sqrt{t + 10d_{\max}+d_{\max}^2/(K^{1/3} \log(K))^2}, \gamma_t^{-1} = \sqrt{\sum_{s=1}^t \sigma_s/\log K + 24^2 d_{\max}^2 K^{2/3}}$, where $d_{\max}$ is the maximum feedback delay and $\sigma_s$ is the number of feedback that are still missing at round $s$. The parameter $\sigma_s$ is essentially used to measure the deviation of the loss estimator in the delayed setting from the ideal no-delay case. In our framework, the proposed parameter $C_t^P$ play an analogous role , derived from an upper bound on the network consensus error and quantifying how far each agent is from the ideal scenario with fully aggregated feedback from all agents. We accordingly choose the learning rates as $\eta_t^{-1} = 4\sqrt{Vt + 169V^2D}$ and $\gamma_t^{-1} = 8V\sqrt{C_t^P t/\log(K) + 36D^2(K-1)^{2/3} + 4(C_t^P)^2}$ for the federated bandits.
>
> > Communication cost: How is $x_{v,t}(k)$ compared to $12VC_t^P\gamma_t$ in (4)? In practice, how many rounds do we need to truncate the loss and broadcast?
>
> In the revision, we slightly increase the learning rate $\gamma_t$ from $\gamma_t^{-1} = 8V\sqrt{(C_t^P t)/\log K + 36D^2 (K-1)^{\frac{2}{3}}}$ to $\gamma_t^{-1} = 8V\sqrt{(C_t^P t)/\log K + 36D^2 (K-1)^{\frac{2}{3}} + 4(C_t^P)^2}$ to reduce the expected communication cost from $O(K+VD)$ to $O(K)$. With this learning rate, the expected total number of rounds with truncation is $O(\sqrt{T})$. This change of learning rate only introduces an additional additive term $O(C_T^P \log K)$ in the regret bound. We have updated our Theorem 1 and its proofs. The proof for the communication round is provided in section 11.4 in the appendix.
>
> We also empirically investigate number of rounds to truncate the loss and broadcast. We reuse the experimental setup in Section 6 for the synthetic and MovieLens datasets. All experiments are repeated for $50$ trials, and we report the averaged results below.
> | Dataset              | Graph    | Diameter | truncation rounds |
> |----------------------|----------|----------|-------------------|
> | Synthetic (Sec. 6.1) | Complete | 1        | 15 rounds         |
> | Synthetic (Sec. 6.1) | RGG-0.5  | 3        | 17 rounds         |
> | Synthetic (Sec. 6.1) | GRID     | 6        | 24 rounds         |
> | MovieLens (Sec. 6.2) | Complete | 1        | 78 rounds         |
> | MovieLens (Sec. 6.2) | RGG-0.5  | 3        | 83 rounds         |
> | MovieLens (Sec. 6.2) | GRID     | 124      | 301 rounds        |
>
> > equation (5): It seems $V$ multiplies to the loss after communication as well?
>
> We apologize for the typo. The cumulative loss estimator update equation should be $\hat L_{v,t+1}^{obs} = \sum_{u:\,(u,v)\in E} P_{u,v}\hat L_{u,t}^{obs} + V\tilde \ell_{v,t}$. We have fixed this typo in the revised version.
>
> > Clarification on matching bounds: If the graph $G$ is known, could we always construct a doubly stochastic matrix $P$ as in Remark 1 to achieve the nearly matching bounds?
>
> The answer is yes. If the graph $G$ is known, we can obtain the degree matrix and the adjacency matrix of the graph. So we can always construct a doubly stochastic matrix $P$ as in Remark 1.
>
> **References:**
> * Koloskova, Anastasia, Sebastian Stich, and Martin Jaggi. "Decentralized stochastic optimization and gossip algorithms with compressed communication." International conference on machine learning. PMLR, 2019.
> * Yang, Shuoguang, Xuezhou Zhang, and Mengdi Wang. "Decentralized gossip-based stochastic bilevel optimization over communication networks." Advances in neural information processing systems 35 (2022): 238-252.

---

### Official Review · Reviewer_39L1 · 2025-11-03

**Soundness:** 3
**Presentation:** 3
**Contribution:** 3
**Rating:** 8
**Confidence:** 3

**Summary:**

This paper considers a multi-agent multi-armed bandit problem, where $V$ agents each face an identical copy of a set of $K$ arms. At each round, agents can share information with their neighbors according to a communication graph. Due to privacy constraints, agents are only allowed to share statistics of the losses rather than the raw losses themselves. The authors design a best-of-both-worlds (BOBW) algorithm that achieves near-optimal regrets in both stochastic and adversarial regimes. The paper also includes numerical evaluations of the proposed algorithm.

**Strengths:**

- This paper establishes new state-of-the-art BOBW regrets for the multi-agent privacy-preserving bandit setting.
- The work introduces a truncated loss estimator, which ensures that individual regrets across agents remain similar (Lemma 1). This is a notable contribution, as prior multi-agent bandit works typically require agents to be fully homogeneous to derive individual regret guarantees.

**Weaknesses:**

- Main concern: The definition of the feedback  $\ell_{v, t}(k_{v, t})$ is unclear. The paper highlights that it is biased, but does not clearly describe the nature or extent of the bias. Moreover, in Section 6.1, the feedback does not appear to show any bias. Could the authors clarify this point?
- It is unclear whether the goal of each agent is to minimize its own individual regret or a global regret across all agents. While the abstract emphasizes that no single agent can identify the globally best arm, this distinction is not explicitly modeled in the problem formulation. Additionally, if each agent is minimizing individual regret, identifying the global best arm may not be necessary. Clarification here would be helpful.
- The use of the term "federated" may be misleading, as federated learning typically involves a central server, whereas the setup in this work appears to be fully distributed.

**Questions:**

- Is the proof of Lemma 2 in the appendix actually meant to support Lemma 1? Please add clear references in the main text to help readers locate the corresponding proofs.
- Why was IND-FTRL not included in the evaluation shown in Figure 2?
- This work achieves a significant improvement in the regret bound for the adversarial regime. Could the authors elaborate on which algorithmic components, analysis techniques, or assumptions are responsible for this improvement compared to (Yi & Vojnović, 2023)?

---

> ### Author Response · Authors · 2025-11-26
>
> Thank you for the insightful review. Here are our response to your comments.
>
> > Main concern: The definition of the feedback $\ell_{v,t}(k_{v,t})$ is unclear.
>
> Thank you for the suggestions. We have added a detailed description of the heterogeneous feedback in Section 3 (lines 159–163). Specifically, different agents may select the same arm at the same time step but receive different loss feedback, and our goal is to identify the globally best arm in hindsight, i.e., the arm whose cumulative loss averaged across all agents is minimal. In the adversarial regime, for each round $t$ and agent $v$, the losses $\\{\ell_{v,t}(k)\\}\_{k\in[K]}$ are arbitrarily chosen by an adversary before the game starts and may vary across agents even for the same arm. In the stochastic regime, for each agent–arm pair $(v,k)$, the sequence $\\{\ell_{v,t}(k)\\}\_{t=1}^T$ is drawn i.i.d. over time from an unknown fixed distribution with mean $\mu_{v,k}$, which may differ across agents.
>
> In Section 6.1, we also provide details on the generation of heterogeneous losses. For each agent $v$ and each arm $k$, we first sample a mean loss $\mu_{v,k}$ independently from Uniform(0, 1). When agent $v$ pulls arm $k$ at round $t$, its feedback $\ell_{v,t}(k)$ is then drawn from a Gaussian distribution with mean $\mu_{v,k}$ and variance $0.01$.
>
> > It is unclear whether the goal of each agent is to minimize its own individual regret or a global regret across all agents.
>
> In our paper, the definition of individual regret (in line 165) consider the average loss across all agents, i.e., $R_T(v) = \mathbb E[\sum_{t=1}^T \bar \ell_t(k_{v,t})] - \min_{k \in [K]}\mathbb E[\sum_{t=1}^T \bar \ell_t(k)]$, where $\bar\ell_{t}(k) $ is the average loss of arm $k$ over all agents at round $t$. Consequently, each agent is minimizing the global regret (although we refer to it as individual regret in the paper) rather than a local regret defined by its own feedback. Thus it is necessary to identify the globally best arm.
>
> > The use of the term "federated" may be misleading, as federated learning typically involves a central server, whereas the setup in this work appears to be fully distributed.
>
> We follow the terminology from prior work on "federated bandits" (Zhu et al., 2021; Yi & Vojnović, 2023; Zhang et al., 2025), all of which consider decentralized environments.
>
> > Is the proof of Lemma 2 in the appendix actually meant to support Lemma 1? Please add clear references in the main text to help readers locate the corresponding proofs.
>
> Thank you for the suggestions. We have reordered the proofs of lemmas in the appendix to improve readability. Now the proof of Lemma 1 is presented in section 10 in the appendix.
>
> > Why was IND-FTRL not included in the evaluation shown in Figure 2?
>
> In the revised version, we have also included IND-FTRL in Figure 2. Since IND-FTRL does not involve any communication, it suffers linear regret, consistent with the behavior already visible in Figure 1.
>
> > This work achieves a significant improvement in the regret bound for the adversarial regime. Could the authors elaborate on which algorithmic components, analysis techniques, or assumptions are responsible for this improvement compared to (Yi & Vojnović, 2023)?
>
> Our improvement mainly comes from the following techniques:
>
> * Note that decentralized communication may introduce latency, as agents can only exchange information with their neighbors. We treat this latency as a form of feedback delay and introduce a novel time-varying parameter $C_T^P$ to quantify the delay induced by decentralized communication.
>
> * To address the issue of locally biased loss feedback, we employ a novel truncated loss estimator that stabilizes per-round updates and keeps the agents’ action distributions nearly aligned, while ensuring that the aggregated loss estimate at each round remains close to the average loss. In addition, we introduce a deviation-record correction scheme that compensates for the bias introduced by truncation, thereby further reducing regret.
>
> * Instead of directly upper-bounding individual regret as in Yi & Vojnović (2023), we first establish an upper bound on the group regret with respect to the global optimum. Leveraging the property that the agents’ action probabilities are nearly aligned, we then approximately divide this group regret by the number of agents to obtain the upper bound of  individual regret. This yields a more fine-grained and tight result.

---

### Author Response · Authors · 2025-11-26

We thank all reviewers for the insightful comments. Regarding the weaknesses and questions raised by the reviewers, we have addressed these concerns in our response and revised the paper accordingly. **All updates are highlighted in blue** in the revised manuscript for ease of reference. The main revisions are summarized below:

* Following the comment of Reviewer 39L1, we now give a clearer description of how heterogeneous feedback is modeled in both the problem setup (Lines 159–163) and the experimental setup (Lines 412–414). We also include IND-FTRL in Figure 2.
* To address the communication cost concerns raised by reviewers cGPZ, yECc, and ouUu, we modified the learning rate $\gamma_t$ to reduce the expected communication cost from $O(K+VD)$ to $O(K)$. We have updated Theorem 1 and its corresponding proofs accordingly.
* Following the reviewer yECc's suggestions, we conducted additional experiments on the robustness of parameter $C_t^P$, which are reported in Section 12.

We believe these revisions improve the overall quality and clarity of the paper, and we remain open to further suggestions during the rebuttal process. Thank you again for your time and constructive feedback during the review process.

---

### Author Response · Authors · 2025-12-02
**Rebuttal Summary**

Dear AC,

We are grateful to the reviewers for their time and effort in evaluating our submission. All reviewers acknowledge that our paper provides strong theoretical guarantees, offers the first Best-of-Both-Worlds regret guarantee for the federated bandits, and significantly improves prior theoretical results in adversarial regime.

At the rebuttal stage, we addressed the reviewers’ concerns in our response and revised the paper accordingly. All updates are highlighted in blue in the revised manuscript for ease of reference. Overall, the only technical concern raised by the negative reviewers (ouUu and yECc) is the  $O(K+VD)$ per-round communication cost. In the revision, we address this issue by modifying the learning rate $\eta_t$, which effectively **reduces the communication cost from $O(K+VD)$ to $O(K)$** while preserving our theoretical guarantees. Reviewer ouUu said that she/he may significantly revise my review comments according to our response. However, as the discussion period was cut short, we did not receive any further feedback.

We summarize our response to the main concerns as follows:

**1. Reviewer 39L1 (Q1) commented that the definition of the feedback $\ell_{v,t}(k_{v,t})$ is unclear.**

We have added a detailed description of how heterogeneous feedback is modeled in both the problem setup (lines 159–163) and the experimental setup (lines 412–414). Specifically, different agents may select the same arm at the same time step but receive different loss feedback $\ell_{v,t}(k_{v,t})$, and our goal is to identify the globally best arm in hindsight.

**2. Reviewers 39L1 (Q6) and yECc (Q3) did not aware which concrete innovations in our algorithm and analysis drive the improved regret bounds compared to prior work.**

Our improvement mainly comes from the following techniques:

* Note that decentralized communication may introduce latency, as agents can only exchange information with their neighbors. We treat this latency as a form of feedback delay and introduce a novel time-varying parameter $C_T^P$ to quantify the delay induced by decentralized communication.

* To address the issue of locally biased loss feedback, we employ a novel truncated loss estimator that stabilizes per-round updates and keeps the agents’ action distributions nearly aligned, while ensuring that the aggregated loss estimate at each round remains close to the average loss. In addition, we introduce a deviation-record correction scheme that compensates for the bias introduced by truncation, thereby further reducing regret.

* Instead of directly upper-bounding individual regret as in Yi & Vojnović (2023), we first establish an upper bound on the group regret with respect to the global optimum. Leveraging the property that the agents’ action probabilities are nearly aligned, we then approximately divide this group regret by the number of agents to obtain the upper bound of  individual regret. This yields a more fine-grained and tight result.

**3. Reviewers cGPZ (Q3), yECc (Q1, Q4) and ouUu (Q1) raised concerns about the per-round communication cost being $O(K+VD)$, viewing it as a key weakness of our approach.**

In the revision, we slightly modify the learning rate $\gamma_t$ from $\gamma_t^{-1} = 8V\sqrt{(C_t^P t)/\log K + 36D^2 (K-1)^{\frac{2}{3}}}$ to $\gamma_t^{-1} = 8V\sqrt{(C_t^P t)/\log K + 36D^2 (K-1)^{\frac{2}{3}} + 4(C_t^P)^2}$, which **reduce the expected communication cost from $O(K+VD)$ to $O(K)$**. With this learning rate, the expected total number of rounds with truncation is $O(\sqrt{T})$. This change of learning rate only introduces an additional additive term $O(C_T^P \log K)$ in the regret bound. We have updated our Theorem 1 and its proofs. The proof for the communication-cost is provided in section 11.4 in appendix.

**4. Reviewer yECc (Q6) asked about the practical robustness of the parameter $C_t^P$.**

We have added experiments to examine the robustness w.r.t. $C_t^P$, presented in Section 12 in appendix. The results show that our FedFTRL algorithm is robust to the choice of the topology parameter $C_t^P$.

**5. Reviewer ouUu (Q2) thought that, with an additional $O(VD)$ budget per round, the problem essentially reduces to a standard MAB with delayed feedback.**

We point out that this is a misunderstanding. Even with an additional $O(VD)$ budget, the studied problem does not reduces to a standard MAB with delayed feedback, as the gossip-based communication always introduces a consensus error. Consequently, each agent cannot obtain the exact global feedback required in standard MAB settings.

---

### Meta-Review · Area_Chair_Hkd5 · 2026-01-05

**Summary:**

The reviewers recognize the paper’s theoretical contribution, improving the regret guarantees for decentralized federated learning bandits, especially the adversarial regret compared to prior work. The primary concerns centered on the communication overhead compared to the existing literature and the novelty of the methodology, given the existing literature on bandits, especially those with delayed feedback. The reviewers have also raised questions about learning-rate design, topology dependence, and robustness to parameter choices.

The authors in the rebuttals have addressed the concerns about the communication overload by reducing to what's already reported in the literature at the expense of additional (but vanishing) regret. They have also provided more discussions on the novelty in methodology. In particular, they have explained that the setting does not reduce to standard delayed-feedback bandits, since gossip-based communication induces unavoidable consensus error.

Since there have not been additional follow-ups by the reviewers and the is a large degree of variability among the ratings, I have carefully read the paper to understand the contribution and roots of the reviewers' comments. I believe the paper makes a meaningful and novel enough theoretical contribution to federated bandits and online learning theory, especially since it is the first work to achieve near-optimal best-of-both-world guarantees in decentralized settings with heterogeneous feedback, and it closes a clear gap in the adversarial regime by improving prior results.

**Reviewer Concerns:**

I believe all the concerns have been addressed sufficiently through a combination of additional theoretical results, more empirical evaluations, and clarification of the methodology introduced/adopted.

**Reviewer Scores:**

It is difficult to comment on that, especially since the most critical reviewer's comments are brief and lack sufficient substance to justify why they believe the technical contributions are not novel. I would imagine that the responses provided addressed their core concern.

---

### Decision · Program_Chairs · 2026-01-26

Accept (Poster)